# Detecting Individual Decision-Making Style: Exploring Behavioral Stylometry in Chess

**Reid McIlroy-Young**
University of Toronto
reidmcy@cs.toronto.edu

**Russell Wang**
University of Toronto
russell@cs.toronto.edu

**Siddhartha Sen**
Microsoft Research
sidsen@microsoft.com

**Jon Kleinberg**
Cornell University
kleinberg@cornell.edu

**Ashton Anderson**
University of Toronto
ashton@cs.toronto.edu

## Abstract

The advent of machine learning models that surpass human decision-making ability in complex domains has initiated a movement towards building AI systems that interact with humans. Many building blocks are essential for this activity, with a central one being the algorithmic characterization of human behavior. While much of the existing work focuses on aggregate human behavior, an important long-range goal is to develop behavioral models that specialize to individual people and can differentiate among them.

To formalize this process, we study the problem of *behavioral stylometry*, in which the task is to identify a decision-maker from their decisions alone. We present a transformer-based approach to behavioral stylometry in the context of chess, where one attempts to identify the player who played a set of games. Our method operates in a few-shot classification framework, and can correctly identify a player from among thousands of candidate players with 98% accuracy given only 100 labeled games. Even when trained on amateur play, our method generalises to out-of-distribution samples of Grandmaster players, despite the dramatic differences between amateur and world-class players. Finally, we consider more broadly what our resulting embeddings reveal about human style in chess, as well as the potential ethical implications of powerful methods for identifying individuals from behavioral data.

## 1 Introduction

The advent of machine learning models that surpass human decision-making ability in complex domains raises the prospect of building AI systems that can collaborate and interact productively with humans. While many advances will be essential in order to achieve this far-reaching goal, algorithmically characterizing aspects of human behavior will be a significant step. Whereas existing work focuses on characterizing human behavior in an aggregate or generic sense, genuine human-AI interaction will require specializing these characterizations to individual people.

Developing algorithms that can detect signatures of individual decision-making style would constitute an important advance for many reasons. First, being able to recognize an individual from their behavioural characteristics would greatly benefit an AI system's ability to coordinate with that individual. In general, decision-making styles can range from complementary to clashing, and a predictive characterization of an individual's decisions would be useful for an AI system trying to adapt to their style. Second, such fine-grained, individual-level characterizations would help isolate the distinguishing features of individual styles. In a domain where ML systems are able to match

35th Conference on Neural Information Processing Systems (NeurIPS 2021).

or surpass human performance, it becomes natural to try extending these systems to pinpoint the types of decisions a person is particularly good at, and types of decisions they could potentially improve. Finally, applying these algorithms to many different individuals would shed light on the structure of decision-making in the domain itself. For example, certain instances could prove particularly discriminative in detecting individual style. In addition to these benefits, the ability to identify individuals from behavioral traces also raises ethical concerns centered around privacy and generalization to high-stakes domains. We discuss the ethical consequences of our work in Section 5.

To formalize the above process, we study the problem of *behavioral stylometry*, in which the goal is to identify a decision-maker from their decisions alone. We pursue this task in the domain of chess, which for many reasons is an ideal model system for developing human-compatible algorithms. AI systems surpassed all human-level ability in chess at least 15 years ago. Despite this, millions of people at all skill levels still actively play. Through their activity, people generate detailed traces of their decisions, which are digitally recorded and publicly available. In chess, behavioral stylometry reduces to predicting the identity of a player from the moves they played in a given set of games.

We present a transformer-based approach to behavioral stylometry in chess. We show how to adapt recent work in speech verification to the problem of detecting individual decision-making style. Working in the domain of chess requires us to extend these methods to the complex types of data found in the game-playing structure of chess. We apply our model in a few-shot classification setup, where only a small number of games are provided for the model to learn a representation of a player's decisions, and the model must then identify the player from thousands of possible candidates, based on another small sample of their games. Compared to a recent approach, our method achieves high accuracy on this task, while being much more scalable and data-efficient. Our resulting embeddings of players and their games reveals a structure of human style in chess, with players who are stylistically similar clustering together in the space. We consider the implications of these findings, including the potential ethical implications of identifying individuals from their behavioral data.

## 2 Related Work

**Speaker verification.** Our work adapts recent work in speaker verification [46, 52], where given a speaker's known utterances, the task is to verify whether an utterance belongs to that speaker. Recent approaches embed utterances into a vector space such that different speakers are distinguishable [28, 42]. These embedding approaches can also be used for downstream tasks such as voice cloning and multi-speaker speech synthesis [21, 27, 34].

A key decision for our task is the choice of loss function, of which two families are most relevant [13]. Classification-based objectives are designed to identify (classify) an unknown target speaker from a pool of possible candidates. Cross-entropy loss is a simple approach, with many variations including angular softmax loss [29], cosine loss [50], and additive angular margin loss [14]. Contrast or metric-learning based objectives instead directly optimize the contrastiveness for samples in different classes, and similarity for those within the same class. Contrastive loss [11], triplet loss [38], and generalized end-to-end loss (GE2E) [49] are used in face verification, face clustering, and speaker verification respectively. In this work, we adapt GE2E to the problem of behavioral stylometry.

**Stylometry.** Both speaker verification and handwriting recognition are stylometry tasks where the goal is to detect and analyze authorial style. Stylometry has historically been associated with attribution studies [45] and has been applied to de-anonymizing programmers from their code [8], identifying age and gender from blog posts [22], and distinguishing machine-generated text from human-generated text [5]. It is often framed as a task to determine the similarity between samples or features extracted from the samples [32]. Handwriting recognition has employed neural methods to identify individuals for decades [7], and development is ongoing [23, 24]. We introduce the task of behavioral stylometry, in which the goal is to identify decision-makers from their decisions.

**Chess.** Chess has a long history in academic research as both a test bed for artificial intelligence [39] and for understanding human behaviour [1, 6, 9]. Recently, these two goals have begun to merge, as researchers have begun to create chess engines that behave in a more human-like manner [30, 37]. Recent work presents personalized chess engines based on AlphaZero [40] that are trained to predict

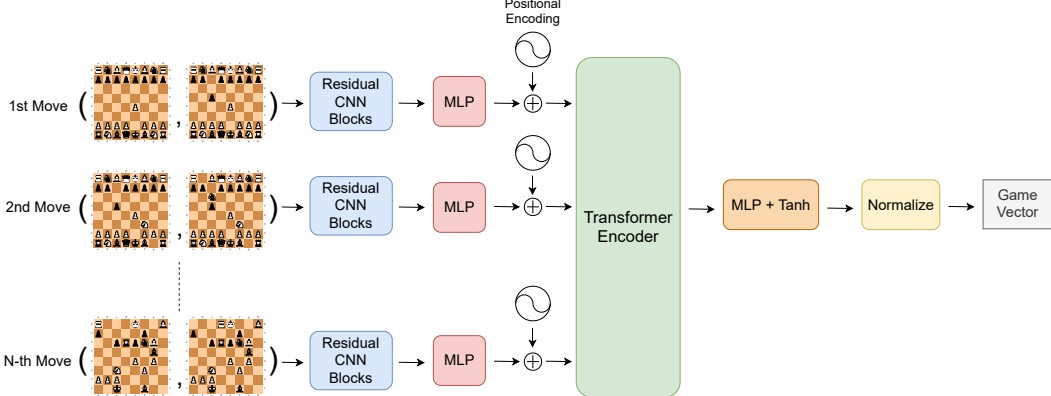

Figure 1: Overview of our model's inference procedure. It first takes in a sequence of chess moves and extracts move-level features with a series of residual blocks, then produces a game vector with a transformer encoder followed by projection layers and normalization.

moves made by specific players [31]. As a side effect, these models can perform behavioral stylometry in chess. We develop a methodology that is both more scalable and more accurate than this approach.

**Few-shot learning.** We approach behavioral stylometry as a few-shot learning task with tens to hundreds of samples per class, which is a well-studied problem formulation [18, 19, 51]. Transfer learning is a main approach to few-shot learning, where models are first trained on a large dataset and then specialized to several smaller datasets [16, 33, 43, 53]. Embedding learning [26] is another approach, where an embedding space is constructed so that similar training samples are close together and different samples are further apart. Test samples can then be mapped to this space and compared with the closest training samples and their associated classes. Several architectures have been proposed, such as matching networks [4, 10, 48] and prototypical networks [36, 41]. Finally, meta-learning [2, 44] is another popular approach, which uses a two-level learning process to train a model across similar tasks and adapts it to solve new tasks with a small number of training samples [20, 35].

## 3   Methodology

We begin by describing our model design for performing behavioral stylometry in chess. The model has three main components: the first extracts features from individual moves, the second aggregates information from the move level to the game level, and the third aggregates information from the game level to the player level. Given a chess position as input, the first step uses residual blocks to extract move-level features. Then in the second step, we aggregate move-level features into a game-level vector using a transformer-based method. Finally, the third step aggregates a player's game vectors by calculating their centroid, producing a player vector that represents the player's individual playing style.

**Move-level feature extraction.** To extract features from chess boards and moves, we draw upon previous work (AlphaZero and Maia) and use a series of residual blocks. The input to the model is encoded as a 34-channel $8 \times 8$ chess move representation; we represent a move with two board positions, corresponding to the boards before and after the player's move. The first 24 channels each encode a specific type of chess piece (6 for each side in one position, and we encode two positions), and the next 10 channels encode position metadata relevant to chess such as repetition, castling rights, the active player's side, 50-move-rule count, and border information. The output is a 320-dimensional feature vector that characterizes each move. Each residual block consists of convolutional layers, batch normalization, ReLU activations, and squeeze-and-excitation (SE) blocks [25]. On top of the residual blocks we add global average pooling, which summarizes the output from residual blocks into a $1 \times 1 \times 320$ feature map, which we flatten into a 320-dimensional vector. The hyperparameters of the model were chosen to match the Maia architecture: 6 residual blocks with channel size of 64 and a reduction ratio for SE blocks of 8.

**Aggregating information to the game level.** The next component of the model takes in a sequence of move features extracted from the first component and constructs a game representation. The input features correspond to the moves played by the target player in a particular game. The output is a game vector that aggregates information about the move features. Although previous work used an LSTM to perform the analogous task in speaker verification, our experimentation revealed that a transformer [47] worked better for our task. Hence, in our approach we use a modified version of the Vision Transformer (ViT) [17], which was originally proposed as a pure transformer replacement for CNNs in vision tasks, achieving comparable results to state-of-the-art CNN architectures with fewer computational resources. As we consider chess moves as $8 \times 8$ images with 34 channels, we can adapt ViT to fit in our setting.

To do this, we first modified the input to ViT to take in a sequence of move features, instead of fixed-size patches extracted from an image. Both BERT [15] and ViT append a classification token to the input sequence, and the final hidden state of this token is considered as a representation of the entire sequence. However, in our task we achieve better performance if we directly average the output vectors of transformer blocks, so we remove the classification token in the input. As all move features are fed into the transformer simultaneously, we retain the ordering of the chess moves by adding a positional encoding to each move feature. The original ViT architecture includes a learnable 1D position embedding in the input sequence, but it cannot easily accommodate variable length inputs during training. Therefore, we use sine and cosine functions to encode positions, a method proposed in the standard transformer, and we include support for encoding up to 500 chess moves.

As an overview of our model, each 320-dimensional input move feature vector is first projected into 1024 dimensions, and a positional encoding is appended to the resulting vectors before they are fed into the transformer encoder. The encoder has 12 transformer blocks, each composed of an attention layer followed by a feed-forward layer, with layer normalization [3] applied before each layer. There are 8 attention heads, each with 64 dimensions, in each attention layer. The feed-forward layer uses MLP with two layers to project the 1024-dimensional vector into 2048 dimensions, where GELU activation is applied, and then another layer projects it back to 1024 dimensions. Each output vector of the encoder corresponds to an input move feature, and we average these vectors to get a single vector of dimension 1024, which is finally projected with MLP into our final embedding space. This 512-dimensional embedding vector encodes the decisions the player made in the game. At inference time, the player vector is obtained by simply averaging all the game vectors (i.e. the centroid) that belong to the player.

**Loss Function.** In order to capture individual playing style, we adapt *Generalized End-to-End Loss for Speaker Verification (GE2E)* [49] to our setting, which allows us to train the game vectors to maximize both intra-player compactness and inter-player discrepancy in the resulting embedding space. GE2E builds a similarity matrix on a batch of $N \times M$ games, where $N$ is the number of players and $M$ is the number of games per player. For each game from each player, we extract the current 512-dimensional embedding vector $\mathbf{y}_{ji}$ ($i$-th game from the $j$-th player) output by the residual blocks and transformer network. We also find the centroid vector $\mathbf{c}_j$ of a specific target player by averaging all of their game vectors in the batch. However, we remove $\mathbf{y}_{ji}$ when computing the centroid of the true player as suggested in [49], to help avoid trivial solutions.

We then construct a similarity matrix $\mathbf{S}_{ji,k}$ containing the cosine similarities between each game embedding vector and each player centroid (for all players in the batch), which we further scale by two learned parameters $w$ and $b$. For the target player, we calculate the centroid $\mathbf{c}_j^{(-i)} = \frac{1}{M-1} \sum_{m \neq i}^{M} \mathbf{y}_{jm}$, and otherwise simply use the average of game vectors for all other players ($k \neq j$).

$$\mathbf{S}_{ji,k} = \begin{cases} w \cdot \cos(\mathbf{y}_{ji}, \mathbf{c}_j^{(-i)}) + b & \text{if} \quad k = j; \\ w \cdot \cos(\mathbf{y}_{ji}, \mathbf{c}_k) + b & \text{otherwise.} \end{cases} \tag{1}$$

The final loss function is designed to maximize the similarity for the correct entries, while penalizing all other entries in the similarity matrix. As a result, the loss function $L(\mathbf{y}_{ji}) = -\mathbf{S}_{ji,j} + \log \sum_{k=1}^{N} \exp(\mathbf{S}_{ji,k})$ pushes each embedding vector closer towards the player's centroid and further from all other players' centroids simultaneously.

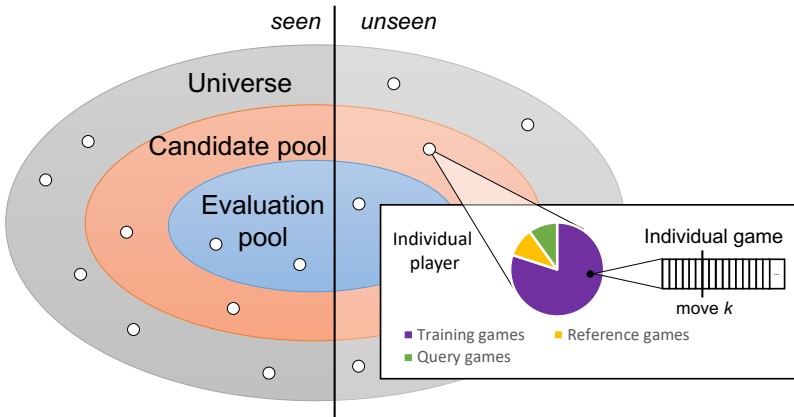

Figure 2: A schematic depiction of how players are organized in a behavioral stylometry task; how each player's games are organized into training, query, and reference sets; and how each game's moves are in a sequential order.

**Training procedure.** We randomly sample $N \times M$ games as a batch, where $N = 40$ is number of players and $M = 20$ is number of games per player. In order to speed up training and perform batch computing, we randomly sample a 32-move sequence from each game and pad 0s to games with length less than 32 moves. The entire model was trained with 4 Tesla K80 GPUs and SGD optimizer with an initial learning rate of 0.01 and momentum of 0.9. The learning rate was reduced by half every 40K steps. Initial values of $w$ and $b$ for the similarity matrix were chosen to be $(w, b) = (10, -5)$, and we used a smaller gradient scale of 0.01 to match the original GE2E model.

# 4 Evaluation

We now train a model using our methodology and evaluate it against competitive baselines on several behavioral stylometry tasks.

## 4.1 Data

**Lichess.** We use chess games downloaded from Lichess, a popular open-source chess platform. Their game database contains over two billion games played by players ranging from beginners to the current world champion, Magnus Carlsen. From this database, we collect rated Blitz games (3–7 minutes per player per game) played on Lichess between January 2013 and December 2020 by players who match all of the following criteria: active as of December 2020, mean rating between 1000–2000, low variance in Blitz rating, and at least 1,000 games played during the time interval. We also restrict our attention to games with at least 10 moves (20 ply). To investigate whether behavioral stylometry performance varies with number of games played, we bucket the players according to how many games they have played: 1K–5K, 5K–10K, 10K–20K, 20K–30K, 30K–40K, and 40K+ (buckets are hereafter referred to by the minimum of their respective range, e.g. "20K" for the 20K–30K bucket). The union of these sets constitutes our *universe* of players $U$.

Players are further divided into *seen* and *unseen* sets, using the same players as our previous work for those with more than 10K games played [31]. Players are *seen* if their games are used in model training and *unseen* if not. Finally, each player's games are randomly split into training games (80% of their games), reference games (10%), and query games (10%). Training games are only used for model training (and thus only seen players' training games are used); reference games are what is known about each candidate player; and query games are what is given about an unknown target player we are attempting to identify. In all of our evaluations, we only use 100 games per player for both the reference and query sets, although in principle they could vary, and for each player their reference and query sets are non-overlapping: $x_r \cap x_q = \emptyset$. The numbers of players and games in each player set is shown in Table 1.

| Dataset | 1K | 5K | 10K | 20K | 30K | 40K+ | Ranked |
|---|---|---|---|---|---|---|---|
| *Seen* Players | 8,617 | 5,298 | 1,866 | 295 | 86 | 19 | 1,813 |
| *Seen* Players' Games | 14M | 24M | 17M | 4.9M | 2.0M | 0.7M | 9.5M |
| *Unseen* Players | 23,101 | 1,324 | 466 | 72 | 20 | 20 | 451 |
| *Unseen* Players' Games | 4.6M | 264.8K | 93.2K | 14.4K | 4K | 4K | 90.2K |

Table 1: The numbers of players and games used in training (*seen*) and evaluation (*unseen*).

**High-ranking players.** We also assembled a dataset of high-ranking online players from both Lichess and Chess.com, the two most popular online chess platforms. To do so, we took the lists of the top 1,500 players on both sites' leaderboards, downloaded all of their games, and filtered out games with fewer than 10 moves and players with fewer than 950 games. We split the remaining high-ranking players into *seen* (80%) and *unseen* (20%) sets, and their counts are shown in Table 1 (again, we use only 100 reference and 100 query games per player in our evaluations). Note that these games include all time controls and tend towards quicker games than the general population. These sets include many of the best players in the world, including the world champion.

## 4.2 Experimental Setup

**Task formulation.** Our central evaluation task is performing behavioral stylometry in chess, in which we are given a set of *query games* played by a particular player, and the task is to identify them from amongst a pool of candidate players $C$, each of whom is associated with a set of *reference games*.

Formally, we set this up as a few-shot classification task. Within our universe of players $U$, we have an evaluation pool $E \subseteq U$ of players we are trying to identify, and a candidate pool $C \subseteq U$ of players which they are guaranteed to belong to ($E \subseteq C$). Every target player in $E$ is associated with a small set of query games $x_q$, and every candidate player in $C$ is associated with a small set of reference games $x_r$ (the conceptual organization of players, games, and moves is shown in Figure 2). In general, the candidate pool $C$ can be much larger than the evaluation pool $E$—for example, one possible task would be to try to identify some set of $|E| = 10$ unknown players, but we are only promised that they are members of some much larger pool of $|C| = 10,000$ players. Finally, we also specify a task parameter $k$ where all games, both query and reference games, only contain moves starting at move $k$. As we will see, the opening moves in chess can be highly discriminative because many players repeatedly play the same openings. For $k = 0$ we are given entire games, but for, e.g., $k = 15$, we are only given moves 15 and beyond for each game. The behavioral stylometry task is thus fully specified by $C$, $E$, $|x_r|$, $|x_q|$, and $k$: for each player in $E$ we are given $|x_q|$ query games and for each player in $C$ we are given $|x_r|$ reference games, all starting at move $k$. For each set of query games $x_q$ we seek to identify which player in $C$ played them.

**Openings baseline.** Chess games always start from the same initial position, and the opening moves are heavily studied and well-understood. Opening choice shapes the resulting game, and players often study a small set of openings in depth and stick to them. As a result, the opening moves are likely to contain a lot of information about who is playing the game, albeit for the somewhat unsatisfying reason that they are memorized patterns of play rather than new decisions being made. Indeed, when strong anonymous accounts play on online platforms, informed observers often rely on their opening moves to guess who they are. As a strong baseline for comparison, we create an embedding model that uses the opening moves. The model creates *game vectors* by sampling 5 sequential moves by the target player, converts each into a one-hot 4096-dimensional encoding vector (moves that are never used will be 0 for all vectors and do not affect our distance measurements), and adds them together to yield a 5-hot vector encoding of the 5 opening moves. Doing this process for each of the $x_q$ games in the target player's set of unlabeled games yields $x_q$ 5-hot vectors in a 4096-dimensional space, which we then convert into a player vector by simply taking their centroid. These player vectors can then be used in the same manner as those in our main model to perform behavioral stylometry.

**Our model.** We apply our methodology to the behavioral stylometry task. To construct our main model, we trained on all training games by all players in the seen universe—63.7M games by 16,181 players in total. Our model learns player vectors in an embedding space that captures chess style,

into which new, unseen players can be encoded. To perform few-shot stylometry on an unseen player $p$ for some candidate pool $C$, we first infer player representations from the reference set of each player in $C$, then infer $p$'s representation by running the model on $p$'s query set. We then compute the cosine distance between $p$'s query representation and all candidate players' reference representations and return the closest candidate player as $p$'s identity. In all of our evaluations, the reference sets are drawn from players' 10% validation sets and the query sets are drawn from players' 10% testing sets. We report Precision at 1 (P@1) unless otherwise noted. We also trained two additional models, one on high-ranking players and one on the set of players used in our prior work [31], which are discussed in the Supplement.

## 4.3 Behavioral Stylometry

**Stylometry performance.** Our main evaluation is the behavioral stylometry task where $C = 10K \cup 20K \cup 30K \cup 40K+$ (all 2,844 Lichess players with at least 10,000 games played, both seen and unseen, are possible identities); $E = 10K+$ (the 578 unseen Lichess players with 10K–40K+ games are the target players to be identified); $|x_q| = |x_s| = 100$ (reference and query sets are 100 games each); and $k = 15$ (we only consider games starting from move 15, so that memorized openings do not factor into our evaluation of capturing human decision-making style). On this task, our model achieves a P@1 of 0.86, exactly identifying a player from 100 of their games out of 2,844 possible candidates 86% of the time (see Table 2 (top)). We emphasize that this is without access to the opening, which is the most regimented, repeated, and identifiable part of a chess game. In comparison, the baseline with access to the first 5 given moves of each game (i.e. moves 16–20) only achieves a P@1 score of 0.234—far better than random guessing, indicating that specific moves indeed capture a significant amount of information, but far worse than our method. We also tested an alternative baseline that uses all moves of the game, which performs slightly better than the opening baseline, but still much worse than our method (P@1 = 0.277, see Supplement for details). In the Supplement, we also show how performance varies for different choices of the evaluation pool. Both our model and the baseline are robust to the choice of minimum total number of games per player, indicating that strong performance on this task is not contingent on players being very active on the platform.

A powerful property of our method is that it adapts to players it has never seen before. We compare performance on unseen players versus seen players (recall that seen players are those whose training games were used during model training). The difference in performance between players the model explicitly trained on and players it has never seen before is surprisingly small. For example, the model achieves P@1 = 0.86 on unseen players and P@1 = 0.853 on seen players. This suggests that the model has learned a comprehensive space of decision-making style from the seen players, and can leverage this to characterize unseen players.

In Table 2 (bottom), we also show the version of the task where the opening moves are given to the models ($k = 0$). Our model performance increases to near-perfect accuracy, but the openings baseline also increases substantially. This indicates that using the opening moves to perform stylometry—identical moves that players repeat out of memorization—is less interesting than using moves later on during games, when players are making decisions in instances they (and our models) have not seen before. Although our model is still significantly better than the openings baseline in this formulation of the task, we use $k = 15$ for the rest of our analyses.

To gauge whether our performance relies on discriminating between player skill levels, we show a variation of behavioral stylometry limited to subsets of players with similar ratings. Restricting our attention to players with ratings within a narrow range around 1900 yields P@1 = 92.6% (see Supplement Table 7 for full details).

**Comparison with personalized move-matching models.** The previous state-of-the-art performance for chess player stylometry was achieved by training individual move-matching models for every player in $C$ [31]. This requires thousands of labeled games per player, and is only computationally feasible for small $|C|$, hence we are unable to include this method in our main analyses due to its lack of scalability. However, we directly compare our method on the small, 400-player candidate pool tested in [31]. For this comparison, the evaluation and candidate pools are both 400 players, $k = 15$, and $|x_q| = |x_r| = 100$. The results are shown in Table 3(top), where personalized models

| Evaluation Pool ($E$) | $|E|$ | $|C|$ | $k$ | Random | Baseline | Our Model |
|---|---|---|---|---|---|---|
| 10k+ Lichess (unseen) | 578 | 2844 | 15 | 0.0004 | 0.244 | **0.860** |
| 10k+ Lichess (seen) | 2266 | 2844 | 15 | 0.0004 | 0.274 | **0.853** |
| 10k+ Lichess (all) | 2844 | 2844 | 15 | 0.0004 | 0.268 | **0.854** |
| 10k+ Lichess (unseen) | 578 | 2844 | 0 | 0.0004 | 0.929 | **0.979** |
| 10k+ Lichess (seen) | 2266 | 2844 | 0 | 0.0004 | 0.930 | **0.982** |
| 10k+ Lichess (all) | 2844 | 2844 | 0 | 0.0004 | 0.929 | **0.982** |

Table 2: Behavioral stylometry performance on Lichess players with all games starting at move $k = 15$ (top) or $k = 0$ (bottom). The candidate pool is the entire set of Lichess players with over 10K games, and the evaluation pool consists of all players that were seen, unseen, or both. Each players' reference and query sets consisted of 100 games each.

| Task | $|E|$ | $|C|$ | Baseline | Personalized | Our Model |
|---|---|---|---|---|---|
| McIlroy-Young et al. | 400 | 400 | 0.478 | 0.552 | **0.953** |
| High-ranked players | 408 | 2157 | 0.039 | — | **0.308** |
| " $\cup$ 10K–40K+ in $C$ | 408 | 5001 | 0.027 | — | **0.301** |

Table 3: (Top) Comparison with personalized move-prediction models in [31] with $k = 15$. (Bottom) Performance on high-ranked players, either with high-ranked players only in the candidate pool or high-ranked players and Lichess amateurs together, with $k = 15$ and $|x_q| = |x_r| = 100$.

achieve P@1 of 0.552. However, our model achieves P@1 of 0.953, far outstripping the previous method's performance.

**High-ranking players.** To examine how our model performs on out-of-distribution examples, we evaluated its performance on our set of high-ranking players (Section 4.1). These players are not only unseen by the model, but they are of much higher skill than any of the players it was trained on, and half of them are drawn from a separate online platform. We begin with the task of identifying the 408 unseen high-ranked players out of the complete set of both seen and unseen high-ranked players (Table 3 (bottom)). This task is significantly more difficult: both the baseline and our model have lower P@1 scores. However, our model still outperforms the baseline by almost a factor of 10. Interestingly, when the set of Lichess amateurs is added to the candidate pool $C$, the model performance hardly changes at all, indicating that high-ranked players occupy a different part of the embedding space.

**Reference and query set sizes.** We studied how model performance varies with $|x_r|$ and $|x_q|$. As Figure 3a shows, model performance rises sharply for earlier values and starts to saturate around $|x_r| = |x_q| = 50$ games. Even when given a small set of labeled examples, i.e., $|x_r| = 10$, our model performs reasonably well. In comparison, personalized move-prediction models require thousands of games to begin seeing improvements over unpersonalized models [31].

**Expanded dataset.** We examined the model's performance on our complete dataset (1K to 40K+), where all 41,184 players are used in both the candidate pool and evaluation pool. Even in this extensive regime, our model achieves a P@1 of 0.54, while the baseline model achieves P@1 of 0.0849 (see Supplement for details).

## 4.4 Qualitative Analysis

**Attention.** Our use of attention allows us to examine which sections of the game are most useful for identifying individual style. Figure 3b shows the average per-move attention for the main model when applied to 100 games for each player in the 40K+ set. Four different starting moves were examined ($k = 0, 5, 10, 15$) to show how the models handle being given incomplete games. The earlier parts of the game are most informative, with a monotonic decay in attention as games progress.

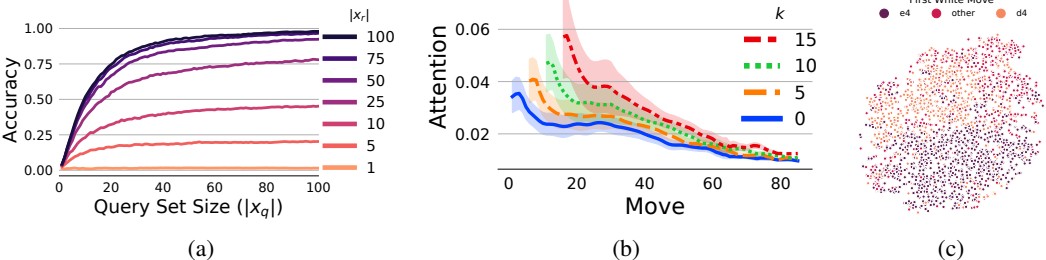

Figure 3: (a) Stylometry performance (P@1) as a function of query set size ($x_q$) and reference set size ($x_r$). (b) Distribution of attention across 1,900 games (40K+ seen players) from 4 different starting moves ($k$). Confidence intervals represent 1 standard deviation. (c) Two-dimensional t-SNE projection of high-ranked game vectors generated from games starting at move $k = 15$, colored by White's first move.

**Playing style.** Figure 3c shows three different colorings of a 2D t-SNE projection of the high-skill players. The colorings are based on the preferred opening moves of the players, when playing as White (see Supplement, Figure 7 for the corresponding projection when playing as black). Despite these player vectors being learned from games starting on move $k = 15$, where the opening moves were discarded, the resulting player embeddings are tightly related to their opening preferences. This indicates that the space is aligned with the stylistic preferences of players.

## 5   Ethical Considerations

As this work pursues the identification of individuals from behavioral traces, it raises important ethical questions. In principle, although our main goal is to specialize behavioral models in the service of more human-compatible AI, the methodology we present and others like it could be abused by bad actors. The most salient concerns revolve around user privacy, as sufficiently powerful stylometry methods could be used to deanonymize individuals. We now discuss these issues in detail, as well as comment on our choice of domain and possible generalization to other domains.

**Privacy concerns.** Our methods can be used to accurately identify people from limited behavioral traces. This raises significant privacy concerns, as well as concerns about power differentials and inequality, since those who are marginalized in society often benefit most from anonymized communication and behavior. The relatively small amount of data required to identify an individual with a high degree of accuracy, at least in the context of chess, would likely surprise those who wish to remain anonymous. Engaged players often have several accounts (either on the same online server or across multiple platforms), and our embedding methodology could be used to link them together. This could be problematic if a player intended to keep an anonymous account separate from a main account, potentially concealing ideas they are experimenting with or details about when they are active online. Similar concerns are also present in the contexts of speech recognition, the domain we adapted our methodology from, and author identification in text, a subfield with a long history.

The history of author identification also makes clear that people have an interest in developing countermeasures to being identified (and again, especially those with limited power). Our work contains insights that can offer help to those seeking such countermeasures in our chosen domain of chess. Consider again a player with multiple accounts. We find that if their accounts were to differ non-trivially in their opening repertoire, this would reduce our model's accuracy in identifying them as the same individual, since we find that the opening phase of the game is the most revealing part. To illustrate, if we divide the games played as White by a single player into two sets, depending on whether they started with the opening move e4 or d4, the resulting set centroids are on average as distant from each other (cosine distance 0.27) as they are from other players' centroids. Of course, any such solution is far from ideal since it requires people to put effort into avoiding identification. But the framework we develop may be able to continue providing further insight into the design of countermeasures to identification requiring limited effort or adaptation from individuals.

Our methods could possibly reveal other information about players through the embedding representations we learn. We do not examine if characteristics such as gender or nationality are correlated with the embedding representations, since the public dataset we use contains no such personally identifiable information, but other machine learning representations have been found to unwittingly incorporate private characteristics [12].

**Domain and generalization.** We chose chess as a first domain to study since it is relatively benign compared to other possible contexts such as human speech, driving behavior, and cellphone mobility data. Having one's chess behavior connected with one's identity, while not always harmless, is arguably less serious than in other domains. It is not a priori clear, in any domain, how uniquely identifying particular forms of behavioral data will be; but it is crucial to understand this since it is easy to build individual-level classifiers from public data of this form, and individuals may act with an expectation that they will remain anonymous, without knowing how revealing their data is. Moreover, there is a quantitative dimension to this—it is not simply all-or-nothing, but a question of how accurately these determinations can be made as a function of the data available. Only through a careful analysis of stylometry can we determine the level of risk in any given domain, and as such it is an important part of making people aware of privacy risks. Our work is following this principle in illuminating the power of stylometry from behavioral traces. By deliberately investigating and quantifying these risks in the domain of chess, we can begin to develop this understanding before something similar is attempted in higher-stakes domains.

The above is particularly important since our methodology can be applied to other potentially more concerning domains. The framework of mapping sequences of decisions into a vector space, and then using an appropriate stylometry loss function, is not unique to chess. We hope that the research community continues to develop both the methodological and ethical frameworks necessary to ensure behavioral stylometry techniques enhance human-AI compatibility and collaboration.

# 6    Conclusion

Motivated by the goal of algorithmically characterizing human behavior, we propose a method for performing behavioral stylometry—identifying people down to the level of specific individuals from their patterns of decisions—and demonstrate the effectiveness of these methods in the domain of chess. We adapt techniques developed for speaker verification tasks to this decision-making setting, and construct a neural network architecture that can reliably identify individual-level decision-making style. Our architecture uses residual blocks to extract position- and move-level features, then combines these with a position encoding and inputs them to a modified Vision Transformer that learns how to accurately aggregate this information into a game-level representation. We use a GE2E loss framework to maximize both intra-player compactness and inter-player discrepancy in the resulting embedding space. The final model is then capable of performing highly accurate few-shot classification of unseen players, outperforming both the previous state of the art and a competitive opening baseline. The resulting embedding space generalizes in several ways—it can identify high-ranked players even though it was only trained on amateurs, it identifies unseen players equally well as players it was trained on, and maintains its strong performance when expanded to tens of thousands of players.

Our methodology has several limitations that raise natural questions for future work. First, while our methodology far outperforms the baseline on high-ranking players, the lower accuracy suggests there is room for improvement in capturing top-class, complex styles. Second, our methodology is geared towards identifying individuals from their behavior alone, which could lead to privacy concerns. While this critique pertains to any stylometry task, including hand-writing recognition and speaker verification, our work raises similar questions in behavioral domains. Future work could investigate whether the benefits of methodologies capable of stylometry could be achieved without the privacy risks of uniquely identifying people. Finally, our methodology could be extended to other behavioral domains where fruitful human-AI collaboration is on the horizon.

**Acknowledgments.**  Supported in part by Microsoft, NSERC, CFI, a Simons Investigator Award, a Vannevar Bush Faculty Fellowship, MURI grant W911NF-19-0217, AFOSR grant FA9550-19-1-0183, ARO grant W911NF19-1-0057, a Simons Collaboration grant, and a grant from the MacArthur Foundation.

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
