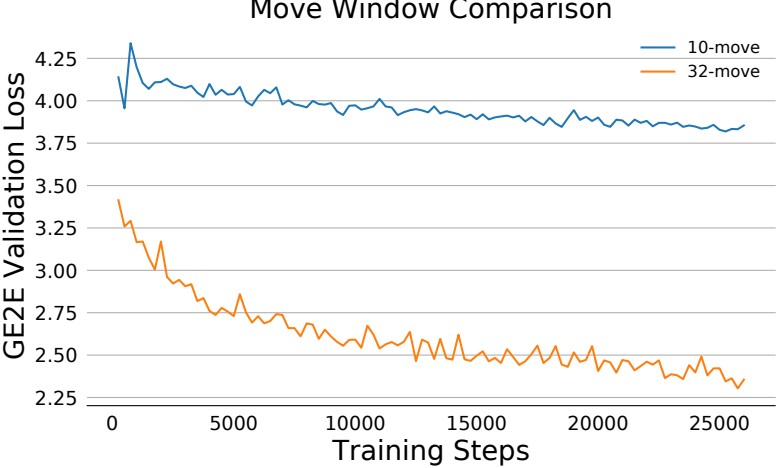

Figure 4: Validation loss when using a 10-move window versus a 32-move window during model training.

# 7 Supplement

This supplement includes additional details about the models discussed in the main text and their stylometry performance, comparisons against other models we trained (e.g., on smaller subsets of the data), and a summary of the artifacts we are releasing.

## 7.1 Training Data Parameters

**Move window.** As mentioned in our methodology (Section 3), we randomly sample a 32-move sequence from each game to speed up the training process. In a given training iteration, this sequence is comprised of moves from one game of one player, but different parts of the game may be sampled during different training iterations. In our earlier experiments, we used a 10-move window since this required no zero-padding to be added to the games (all games have at least 10 moves). However, we later observed a benefit from using longer sequences of the input, even though some games have fewer than 32 moves and require zero-padding. This comparison is shown in Figure 4.

**Starting move.** The models discussed in the main text are trained on any 32-move window from each game. During inference time, we distinguished between the easier task of using the opening moves to perform stylometry (starting at move $k = 0$) versus the harder task of using moves later on during the games (e.g., starting at move $k = 15$), when players are making decisions in instances they (and our models) have not seen before. However, this leaves the question of how opening moves impact *training* unanswered. To investigate this, we additionally train a model using 32-move windows that only start at move 15 or later. The stylometry accuracy of this model is 0.390 on a candidate pool $|C| = 232$ and evaluation pool $|E| = 232$. When evaluated on the same exact task, our main model (which does include the opening moves in its training data) achieves a significantly higher accuracy of 0.9927. This reveals that information about the opening moves is quite important during model training: our main model likely draws on this information when evaluated on later moves, whereas the model trained only on moves 15 or later does not have access to such information directly.

## 7.2 Model Architecture Variants

**LSTM vs Transformer.** In our initial experiments, we used the same LSTM network that was used in the original GE2E work [49]. We subsequently observed a small performance gain when using a Bi-directional LSTM. This is likely due to the transformer being better able to focus on the early game as show in figure 3b. Figure 5 shows that a Transformer model consistently outperforms a

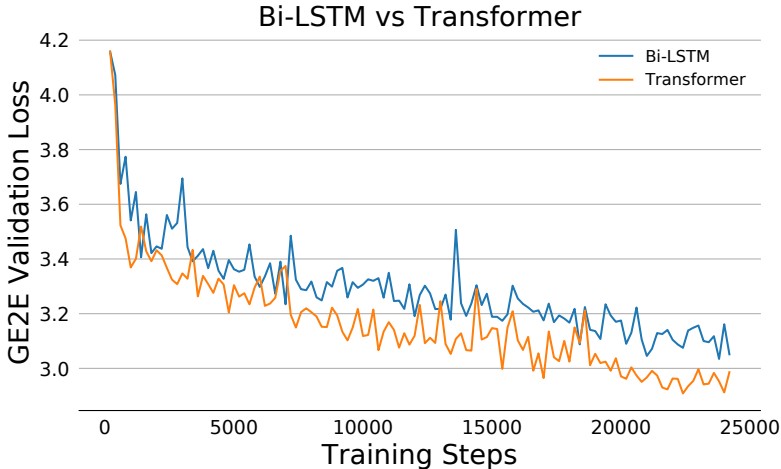

Figure 5: Validation loss of a Bi-directional LSTM compared to the Transformer model.

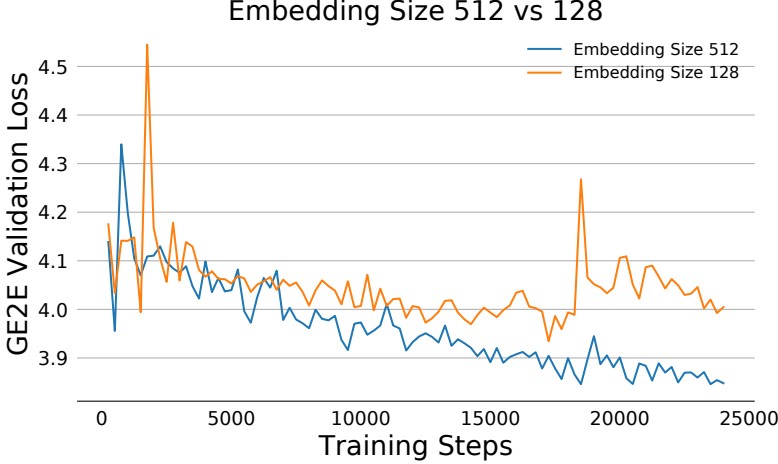

Figure 6: Validation loss when using a 512 vs. 128-dimension embedding.

Bidirectional LSTM throughout the training process. Table 6 also shows the LSTM's accuracy when fully trained.

**ArcFace vs GE2E.** As we discussed in related work (Section 2), there are two families of loss functions that are most relevant to our behavioral stylometry task: ArcFace [14] and GE2E [49]. During our investigation, we experimented with one loss function from each of these families. However, holding everything else constant, we observed that ArcFace consistently failed to converge in the settings we evaluated.

**Embedding size.** The embedding size is also an important hyperparameter that we explored. We experimented with both a 128 and 512-dimensional embedding, as shown in Figure 6. Even though the two models perform similarly during the first 10K training steps, the validation loss for the 512-dimensional model continues to drop beyond that point, while the 128-dimensional model roughly stays the same (Figure 6). All of our models thus use an embedding size of 512.

### 7.3 Number of Games Played

We can partition the players by the number of games they have played to match the partitioning used by [31]. Table 4 shows the accuracy of our main model as we vary the evaluation pool of the number

| Evaluation Pool ($E$) | $|E|$ | $|C|$ | Random | Baseline | Our Model |
|---|---|---|---|---|---|
| 10k − 40k+ | 578 | 2844 | 0.0004 | 0.244 | 0.860 |
| 20k − 40k+ | 112 | 2844 | 0.0004 | 0.286 | 0.848 |
| 30k − 40k+ | 40 | 2844 | 0.0004 | 0.325 | 0.875 |
| 40k+ | 20 | 2844 | 0.0004 | 0.300 | 0.900 |

Table 4: Model accuracy on different sets of Lichess players based on number of games played. Each model is given games starting from the 15th move to create game vectors. The candidate pool is the entire set of Lichess players with over 10K games in our dataset; 100 games from each player's test set is used for both the reference and query sets of each player.

| Evaluation Pool ($E$) | $k$ | $|E|$ | $|C|$ | Baseline | Our Model |
|---|---|---|---|---|---|
| 5k − 40k+ | 15 | 9465 | 9465 | 0.171 | **0.737** |
| 1k − 40k+ | 15 | 41184 | 41184 | 0.0849 | **0.540** |

Table 5: Comparison of baseline and our model on a large dataset of players. Candidate and evaluation pool are the same set and include players from the seen and unseen sets, all games are from the test set. 100 games were used for query and reference sets.

of games played, and use the corresponding set of players in each candidate pool. The full set of seen and unseen players is used for this analysis, but as in the main paper, the testing sets for all players are used for the evaluation

## 7.4   Expanded Dataset Results

The most important results for the large data set are covered in Section 4.3, but we also ran other tests that are summarized in Table 5.

The results for the baseline and our models on the task of identifying players from a large set can be broken down by the player's category. This task is constructed the same way as in the main paper with $k = 15$. Looking at players with 5K or greater games, the baseline achieves 0.171 accuracy while our model achieves 0.737 accuracy. The evaluation and candidate pools are both the same (of size 9465) since we use all players. When we look at the largest possible set of Lichess players—those with at least 1K games—we see 0.0849 accuracy for the baseline and 0.540 for our model.

## 7.5   Alternative Models Comparison

Table 6 shows the accuracy of an alternative baseline model which uses all positions instead of a 5 move window and an LSTM model. The LSTM model uses an LSTM instead of a transformer for the variable input length, and was trained on the same data as the main model. The all moves model while slightly better has higher overhead so for the 41k players test (section 4.3) was infeasible.

## 7.6   Accuracy on Rating Subsets

Table 7 shows the accuracy on sets of players with different ratings. Note that the rating is the minimum and the buckets are of width 100, *e.g* 1100 refers to players rated $[1100, 1200)$.

| Evaluation Pool ($E$) | $|E|$ | $|C|$ | $k$ | Random | All Moves | LSTM | Transformer (Final Model) |
|---|---|---|---|---|---|---|---|
| 10k – 40k+ | 578 | 2844 | 15 | 0.0004 | 0.2768 | 0.7803 | 0.8599 |
| 20k – 40k+ | 112 | 2844 | 15 | 0.0004 | 0.3750 | 0.8036 | 0.8482 |
| 30k – 40k+ | 40 | 2844 | 15 | 0.0004 | 0.4500 | 0.8750 | 0.8750 |
| 40k+ | 20 | 2844 | 15 | 0.0004 | 0.4500 | 0.9000 | 0.9000 |

Table 6: Comparison of alt.ernative models not discussed in the main text

| $k$ | $|E|$ | $|C|$ | Rating | Model Accuracy |
|---|---|---|---|---|
| 15 | 19 | 19 | 1100 | 1.000 |
| 15 | 36 | 36 | 1200 | 0.972 |
| 15 | 54 | 54 | 1300 | 0.926 |
| 15 | 106 | 106 | 1400 | 0.981 |
| 15 | 162 | 162 | 1500 | 0.988 |
| 15 | 297 | 297 | 1600 | 0.976 |
| 15 | 340 | 340 | 1700 | 0.968 |
| 15 | 487 | 487 | 1800 | 0.926 |
| 15 | 522 | 522 | 1900 | 0.925 |

Table 7: Accuracy on sets of players with different ratings. 1100 means players in $[1100, 1200)$, 1200 players in $[1200, 1300)$, . . . .

## 7.7 Training on Other Players

In the main text, we train our model on a large set of amateurs on Lichess (Section 4.2). Here, we report on two additional experiments in which we train on a separate set of interesting players. In the first, we train our model only on the 400 players used in [31]. This is an experiment with training on orders of magnitude fewer players, and also facilitates a direct comparison with previous work (and our main model's performance on the 400 players from previous work). In the second, we train on strong, high-ranking players from both Lichess and Chess.com, the two largest online chess platforms. This is also a smaller training set, and helps us understand whether our training methodology generalizes to high-level players. Both models performed worse on their respective tasks than our main model due to the small training set size, and are included here for comparison and completeness.

| Task | Evaluation Pool | $|E|$ | $|C|$ | Baseline | Our Model | High Model | 400 Model | Model |
|---|---|---|---|---|---|---|---|---|
| 10k – 40k+ | all | 15 | 2844 | 2844 | 0.268 | 0.854 | 0.202 | 0.515 |
| 10k – 40k+ | seen | 15 | 2266 | 2844 | 0.274 | 0.853 | 0.204 | 0.520 |
| 10k – 40k+ | unseen | 15 | 578 | 2844 | 0.244 | 0.860 | 0.195 | 0.497 |
| 20k – 40k+ | seen | 15 | 400 | 2844 | 0.355 | 0.870 | 0.216 | 0.573 |
| 20k – 40k+ | unseen | 15 | 112 | 2844 | 0.286 | 0.848 | 0.273 | 0.607 |
| 30k – 40k+ | seen | 15 | 105 | 2844 | 0.457 | 0.886 | 0.258 | 0.590 |
| 30k – 40k+ | unseen | 15 | 40 | 2844 | 0.325 | 0.875 | 0.282 | 0.725 |
| 40k+ | seen | 15 | 19 | 2844 | 0.368 | 0.895 | 0.294 | 0.737 |
| 40k+ | unseen | 15 | 20 | 2844 | 0.300 | 0.900 | 0.263 | 0.700 |
| High-ranked $\cup$ 10K–40K+ | all | 15 | 4911 | 4911 | 0.158 | 0.595 | 0.136 | 0.325 |
| High-ranked $\cup$ 10K–40K+ | seen | 15 | 3925 | 4911 | 0.160 | 0.590 | 0.137 | 0.327 |
| High-ranked $\cup$ 10K–40K+ | unseen | 15 | 986 | 4911 | 0.151 | 0.614 | 0.130 | 0.318 |
| High-ranked, high only $C$ | seen | 15 | 1659 | 2157 | 0.026 | 0.273 | 0.107 | 0.108 |
| High-ranked, high only $C$ | unseen | 15 | 408 | 2157 | 0.039 | 0.308 | 0.106 | 0.118 |
| High-ranked | seen | 15 | 1659 | 4911 | 0.019 | 0.265 | 0.080 | 0.097 |
| High-ranked | unseen | 15 | 408 | 4911 | 0.027 | 0.301 | 0.081 | 0.108 |
| McIlroy-Young et al. | seen | 15 | 400 | 400 | 0.477 | 0.953 | 0.480 | 0.825 |

Table 8: Behavioral stylometry performance on Lichess and high-ranked players with all games starting at move $k = 15$. The candidate pool is the entire set of Lichess players with over 10K games on the top, and various other combinations as noted on the bottom. The evaluation pool is as described with players in either seen set, unseen set, or both ("all"). 100 games were used for both reference and query sets of each player.

| Task | Evaluation Pool | $|E|$ | $|C|$ | Baseline | Our Model | High Model | 400 Model | Model |
|---|---|---|---|---|---|---|---|---|
| 10k – 40k+ | all | 0 | 2844 | 2844 | 0.930 | 0.982 | 0.455 | 0.810 |
| 10k – 40k+ | seen | 0 | 2266 | 2844 | 0.930 | 0.982 | 0.464 | 0.815 |
| 10k – 40k+ | unseen | 0 | 578 | 2844 | 0.929 | 0.979 | 0.419 | 0.791 |
| 20k – 40k+ | seen | 0 | 400 | 2844 | 0.948 | 0.988 | 0.497 | 0.890 |
| 20k – 40k+ | unseen | 0 | 112 | 2844 | 0.893 | 1.000 | 0.446 | 0.812 |
| 30k – 40k+ | seen | 0 | 105 | 2844 | 0.933 | 1.000 | 0.533 | 0.924 |
| 30k – 40k+ | unseen | 0 | 40 | 2844 | 0.850 | 1.000 | 0.475 | 0.850 |
| 40k+ | seen | 0 | 19 | 2844 | 0.947 | 1.000 | 0.684 | 1.000 |
| 40k+ | unseen | 0 | 20 | 2844 | 0.900 | 1.000 | 0.500 | 0.900 |
| High-ranked $\cup$ 10K–40K+ | all | 0 | 4911 | 4911 | 0.789 | 0.799 | 0.345 | 0.557 |
| High-ranked $\cup$ 10K–40K+ | seen | 0 | 3925 | 4911 | 0.785 | 0.800 | 0.347 | 0.558 |
| High-ranked $\cup$ 10K–40K+ | unseen | 0 | 986 | 4911 | 0.806 | 0.795 | 0.338 | 0.552 |
| High-ranked, high only $C$ | seen | 0 | 1659 | 2157 | 0.614 | 0.569 | 0.283 | 0.259 |
| High-ranked, high only $C$ | unseen | 0 | 408 | 2157 | 0.651 | 0.554 | 0.295 | 0.278 |
| High-ranked | seen | 0 | 1659 | 4911 | 0.605 | 0.565 | 0.243 | 0.244 |
| High-ranked | unseen | 0 | 408 | 4911 | 0.649 | 0.549 | 0.263 | 0.266 |
| McIlroy-Young et al. | seen | 0 | 400 | 400 | 0.990 | 0.995 | 0.740 | 0.973 |

Table 9: Behavioral stylometry performance on Lichess and high ranked players with all games starting at move $k = 0$. The candidate pool is the entire set of Lichess players with over 10K games on the top, and various other combinations as noted on the bottom. The evaluation pool is as described with players in either unseen set, seen set, or both ("all"). 100 games were used for both reference and query sets of each player.

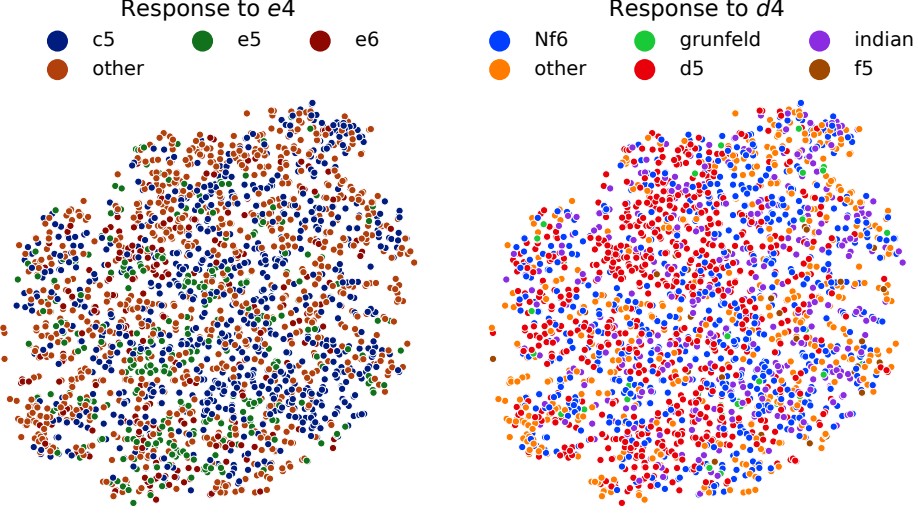

(a) Colored by first move in response to e4.     (b) Colored by first move in response to d4.

Figure 7: Two-dimension t-SNE projection of high-ranked player vectors generated from games starting at move $k = 15$.

## 7.8   Additional Embedding visualizations

Figure 7 shows the same embedding layout as figure 3c but colored by the moves made in response to white's first move.

### 7.8.1   400 Player Model

We train a 400-player model on the same 400 players that [31] uses for its stylometery analysis. In contrast with the main model, which uses 16,000 players for training, this one is drastically reduced to 400 players.

The results for this model are shown in the *400* column of Tables 8 and 9. The main comparison task for this model is the McIlroy-Young et al. dataset which is the same set of players (but not games) it was trained on. It outperforms the baseline on the $k = 15$ tasks (Table 8), but not on the $k = 0$ tasks (Table (table 9)). Additionally, it does not generalize well and performs badly on the high-ranked players task. These results suggest that more than 400 players are needed to learn the space of chess-playing style.

### 7.8.2   High-Ranking Player Model

We also trained a high-ranking player model with the 1,813 high-ranked training players. Having a model that performs well on high-skill players is of interest to the chess community. While our final model presented in the main text performs well above the baseline, we were interested in evaluating whether directly training on high-skill players would perform even better.

The results are shown in the "High" column of Tables 8 and 9. Similarly to the 400 model, it under-performs our final model by a large margin. Notably, the model explicitly trained on high-ranked players performs worse than our main model on the high-ranked players task, suggesting that high-ranked players are a harder set to learn from. This is consistent with our observation that they are harder to distinguish.

### 7.9 Artifacts for Release

#### 7.9.1 Code

Our public code release can be found at `github.com/CSSLab/behavioral-stylometry`. As there are ethical concerns regarding releasing the model and full training code it may not be publicly accessible. You can contact our corresponding author at: `reidmcy@cs.toronto.edu` to request access.

We incorporate three pieces of open source code in ways that are not trivially separated from our own code. One is GPL licensed and handles the board representations, the second and third are MIT licensed. One provides help with the transformer implementation and the other is for the training pipeline. All are noted in this code release.

#### 7.9.2 Data

The data used for training are all from the Lichess database which is free and released under a Creative Commons CC0 license. The code for processing and analysis is included with the code release.

#### 7.9.3 Interactive embedding viewer

We used Embedding Projector to view the embedding vectors. Instructions on how to view it can be found with our code release at `github.com/CSSLab/behavioral-stylometry`