# OpenReview forum: "Detecting Individual Decision-Making Style: Exploring Behavioral Stylometry in Chess"
_NeurIPS.cc/2021/Conference — NeurIPS 2021 Poster_

### Official Review · Reviewer_iTpg · 2021-07-15

**Rating:** 7
**Confidence:** 4

**Summary:**

This paper introduces a transformer-based neural network architecture that can be used to learn and identify unique playing styles of individual chess players. The neural network learns an embedding over players' actions and by using the distance between the learned embedding and a sequence of chess moves, the network can identify the player taking those moves. While this problem has been studied before, the main contribution of the paper is the ability of the architecture to learn the playing styles of unseen players using a small number of example moves.

**Ethical Concerns:**

The work pertains to explicitly de-anonymizing the identities of people based on limited data. The authors do not sufficiently address the negative implications of abuse of such technology.

**Ethics Review Area:**

["Inappropriate Potential Applications & Impact  (e.g., human rights concerns)", "Privacy and Security (e.g., consent)"]

**Limitations And Societal Impact:**

The authors do not sufficiently address the societal impacts of their work. There is one line in the conclusion that addresses this but it is not enough. Especially, given the security and privacy concerns that could arise from misuse of the research. While the work is limited to a relatively harmless domain, there is potential of its use in more sensitive domains. Given that identifying people based incomplete information can be abused in a number of situations, the authors should have spent more time elaborating on the positive/negative implications of their work.

**Main Review:**

The main idea and motivation behind the paper is well defined and I found the problem of few-shot learning of player styles interesting. However, the comparison with a baseline used in the experiments from a previous paper is flawed -- see below.

Originality: While none of the techniques used in this paper are novel, the authors do employ previous techniques to a problem in which they are yet to be used. The use of learned embeddings to characterize player styles was clever.

Clarity: The paper leaves much to be desired in terms of clarity. I think the paper could have been written better. One subsection in which this is apparent is the description of the loss function. The explanation reads as if it structured in reverse --- that is, it would have made more sense if the similarity matrix was described before the embeddings. In fact, the description of the embeddings uses notations not defined until the similarity matrix was explained. I ended up having to refer the to the cited literature in order to understand this part. Furthermore, I think the paper focusses too much on the description of the architecture specifications rather than on the reason this architecture was chosen. Given that the authors claim that the strength of their proposed model lies in the use of transformers, it would have been prudent for the authors to describe why this is so --- why are transformers better? What characteristics of transformers make them more suitable to the task of behavioral stylometry?

Quality: The experimental evaluation has a potentially major issue in regards to the comparisons performed against previous methods. Authors, please clarify this in your response, if I misunderstood the comparison. As far as I can tell, the paper misrepresents its comparative performance. The paper presents behavioral stylometry on settings of k = 0 and k = 15. While the authors state that they compare against previous work at a setting of k = 15, this is not what they do. The results from previous work on an identical problem were performed at settings of k = 0, 10 and 30. The performance of the previous work reported in the paper were for k = 30. The prediction problem is much more difficult with higher k values (as the authors themselves make clear.) This invalidates any comparison with the previous algorithm.

Significance: The paper does not present anything wholly unique, but the results reported are promising (if one discounts the potential major flaw described in the paragraph above). Another limiting factor of the paper's significance is that it uses a single problem domain in its evaluation.

COMMENT AFTER RESPONSE

Thanks for your response. In particular for clarifying the issue with the k-value.

**Needs Ethics Review:**

Yes

**Time Spent Reviewing:**

12

---

> ### Author Response · Authors · 2021-08-11
> **Author Response to Reviewer iTpg**
>
> Thank you for your thoughtful feedback and comments.
>
> We wrote a general statement on all the reviewer concerns regarding privacy/negative implications, which is included with the comments specific to your review.
>
> Additionally you taking the time to check our references is impressive and we thank you for the time you took and we hope your concerns are addressed in our clarification:
>
> ### _k_ value (also `a4DH`)
> In this paper, we use the convention that one move equals an action by both players (1 move = 2 ply). This indexing is natural for our work since our model takes sets of boards as input, but we will make the definition and comparison with previous work more explicit in the final version. Specifically, this means that our results are directly comparable with McIlroy-Young et al., as they conduct analyses on k=0, k=7.5, and k=15 using our indexing system, and we conduct analyses on k=0 and k=15. We will clarify this in the Table 3 caption.
>
> ### Privacy (all)
>
> The reviewers all highlighted the privacy and ethical implications of this work, which we comment on here. We have been engaging with the ethical implications of our work in this domain throughout the process, both in the discriminative sense that the reviewers discuss (the privacy concerns), and in a generative sense (the ability to impersonate players). Because of this, we have worked with our institutional IRB to develop a protocol that has been approved prior to the paper for conducting research on these problems. Given this, we also agree that the discussion of these points in the paper was much too short, and didn’t accurately reflect the extent of ethical considerations that have gone into the research. We would plan to expand on this in the revised version.
>
> We agree that identification of individuals from behavioral traces raises significant privacy concerns, as well as concerns about power differentials and inequality, since it is often people who are marginalized in society who benefit most from anonymized communication and behavior. Taking this into account, it is useful to distinguish between two kinds of research into stylometry and de-anonymization. One form of this research, which we do not engage in, uses information or meta-data that is not generally publicly available to discover identities that would be infeasible to figure out without this type of data. A second form of this research uses data that is public and easily accessible to determine how distinguishable different individuals are from these public representations.
>
> This second form is the genre of stylometry research that we engage in — since all the chess data that we use is easily downloadable, and public by design — and we argue that this type of stylometry is important as part of a research program for understanding and addressing privacy risks in different domains. In particular, it is not a priori clear in any domain how uniquely identifying particular forms of behavioral or content data will be, but it is crucial to understand this, since it is easy to begin building individual-level classifiers from public data of this form, and individuals who operate in this domain with an expectation that they’ll remain anonymous may be relying on unrealistic assumptions about how revealing their data is. Moreover, there is a quantitative dimension to this – it is not simply all-or-nothing, but a question of how accurately these determinations can be made as a function of the data available.
>
> Only through a careful analysis of stylometry can we determine the level of risk in any given domain, and as such it is an important part of making people aware of privacy risks. And when public data is easily accessible, it is crucial to understand these risks. Our work is following this principle in illuminating the power of stylometry from behavioral traces, starting from an understanding of the risks involved. Finally, we note that we deliberately investigate and quantify these risks in the relatively harmless domain of chess, which we believe to be a valuable step before something similar is attempted in higher-stakes domains where more is at risk.
>
> ### Short Point
>
> We will work to improve the clarity of the paper in the final release, specifically improve the methodology section and notation
>
> ### Model Choice
>
> We hypothesized that the use of a transformer architecture would improve performance because the task depends on decisions made throughout the game, with some presumably being much more important than others. The self-attention mechanism is designed to allow the model to use any past state easily, while models such as LSTMs use a hidden state vector to indirectly incorporate past states in their outputs. We also tested the performance of both and found transformers had lower validation error and loss during our model tuning. As an additional test we have trained a bi-directional LSTM model using the full dataset and it achieves 78.0% accuracy on the 10k+ task with k=15, which is lower than the transformer model which achieves 86.0% on the same task. The loss and error on validation set are also worse when compared with a bi-directional LSTM model. The equal error rate for the models was: 0.225 LSTM and 0.187 transformer, and the loss was 2.502 LSTM and 2.323 transformer. Conditional action distribution (qhYv). The reviewer made the useful suggestion of comparing against a naïve baseline using conditional action distributions. In response to this, we implemented the following baseline: for each user, we computde their empirical distribution over the 4,096 possible actions for all their moves in their support set of of games (starting with move k=15 for direct comparison with other models). We then treated these as vectors, as in our main model, and compared their similarity for identifying players. This resulted in an accuracy of 27.7% (compared with 24.4% for our 5-opening-moves baseline) on the 10k+ task. We will include a full analysis in the final paper.
>
> ### Generalization (also `qhYv`)
> We propose chess as a model system in which we introduce the problem of behavioral stylometry and demonstrate that a technique from another domain (human speech synthesis) does very well. Thus the method has generalized from at least one other domain, and we see chess as a natural testbed to explore both how identifiable people are from their actions and the benefits versus risks of behavioral stylometry in a relatively low-risk domain. We hope the insights we gain here will provide insight into applying these methods to other domains in future work.

---

> > ### Comment · Reviewer_iTpg · 2021-08-26
> > **Thank you!**
> >
> > Thank you for your response. I have updated my score accordingly.

---

### Official Review · Reviewer_a4DH · 2021-07-16

**Rating:** 4
**Confidence:** 3

**Summary:**

This paper describes a transformer-based approach to determining chess player identity from moves played.

**Limitations And Societal Impact:**

The paper would benefit from further motivation of the problem domain and further discussion of ethical issues concerning behavioral stylometry. Is chess simply a testbed for general-purpose behavioral stylometry methods, or are there real-world applications of behavioral stylometry in chess? I see two main ethical issues with general-purpose behavioral stylometry, which merit further discussion:
 - Privacy Concerns -- this is briefly mentioned in the conclusion but deserves further discussion and citation of related ethics work. For example, the related work stylometry section mentions face verification, de-anonymization of programmers, and "identification of age and gender from blog posts" but does not mention the ethical issues raised by these application domains. Further, detecting user identity from online behavior carries many ethical risks and the tradeoffs are not well-described in this paper. For example, what are the "benefits of methodologies capable of stylometry" in this context? What are the risks? [Although I think this is mainly an issue in other stylometry domains, I don't particularly want my (anonymous) Lichess account to be connected to my (public) Chessbase data, either.]
 - Performance Ceilings -- high-accuracy stylometry is impossible in many domains, such as author identification. This can have ethical implications for e.g., forensic domains, where it is critical that policymakers and other non-technical individuals understand the limitations of these methods. Even in a domain like chess, methods for cheating detection (which became especially popular during virtual tournaments due to COVID-19) are limited in their accuracy, and accurately conveying that *it is probably impossible to identify players with perfect accuracy* is important for helping non-ML practitioners understand and make use of these results.

**Main Review:**

**Review Criteria**

Originality: McIlroy-Young et al. (2020b) have previously studied behavioral stylometry in chess. This paper provides a more accurate and scalable solution to an existing task domain using transformer-based game embeddings. The architecture is adapted from work on speaker adaptation but is novel in its application to this domain.

Quality: The results appear technically sound and the proposed method achieves high accuracy. The authors run appropriate ablations, such as not including opening moves, evaluating out-of-domain generalization, etc. [These ablations are especially important because the accuracy values are *shockingly* good; as an expert chess player (~ELO 2100), I am quite surprised that it is theoretically possible to identify chess players with such high accuracy, and I was quite skeptical of the results at first.]

Clarity: The task and model architecture are clearly described. Some details of the few-shot classification setup are difficult to understand.

Significance: Behavioral stylometry in chess is not a particularly significant task, but the methods described could easily apply to other domains involving decision-making style, and chess serves as an appropriate and convincing testbed. In particular, the comparison between the embedding method and personalized "move-matching" models is interesting and likely to generalize to other domains. However, as noted below, stylometry is wrought with potential ethical issues that should be discussed further.

**Notes on Specific Sections**

Abstract: First paragraph of the abstract can be removed, since it reads more like an introduction and is in fact repeated in the introduction. I would recommend starting the abstract by describing what you did, rather than motivating the problem domain.

Introduction: It might be worth defining "style" and what it means in the context of chess. For example, one common distinction is between aggressive/tactical vs. positional play, with certain players (Shirov, Tal) being known for aggressive play, and certain openings (e.g., gambits) being more or less tactical in nature. Presumably, this method also picks up on elements of "style" which do not readily correspond to human-interpretable concepts.

Related Work/Methodology: Overall, this is all very clear and well-written. Possibly out-of-scope, but I wonder if you could achieve better results by initializing with board embeddings from the policy network of AlphaZero or some other neural chess engine. [But the results are already quite good, so this isn't top priority.]

Data: I'm not sure why the paper groups by the number of games played when the evaluation uses only 200 games for each player. However, sampling from ELO 1000-2000 is quite a large range, and I'm curious if these results would still hold for a narrower range. It seems possible to me that a model could be evaluating player strength rather than "style." Similarly, the top 1500 players from Chess.com/Lichess will also exhibit a lot variance in strength.

Evaluation:
 - Task formulation: I found this section difficult to read, possibly because I had previously assumed D = R.
 - Can you confirm that k=15 means 15 moves for both players? Normally in chess a "move" consists of plays from both white and black, but I just wanted to confirm that's the case in your paper. There is technically opening theory that extends beyond the 15th move, but especially in Lichess 1000-2000 games, I wouldn't expect it to be very common. However, if k=15 actually meant 15 turns for both players combined (i.e., 7.5 moves), then this would definitely still be within the realm of opening theory, even for many amateur players.
 - Results for k=0/k=15 and varying sizes of support/query sets are useful, thanks!
 - Attention results seem straightforward and confirm my intuitions that openings are the most useful part of the game for stylometry, except for maybe the first move (where there are just less choices to be made). I'm less clear what takeaways I should have from the t-SNE plots, especially 3b and 3c. Does this just mean that choice of first move is an important factor in style but responses or not? Or how should I interpret 3b-3c?

Conclusion: Additional discussion of privacy concerns would be appreciated. See below.

**Time Spent Reviewing:**

3

---

> ### Author Response · Authors · 2021-08-11
> **Author Response to Reviewer a4DH**
>
> Thank you for your thoughtful feedback and comments.
>
> We wrote a general statement on all the reviewer concerns regarding privacy, which is followed by comments specific to your review.
>
> ### Privacy (all)
>
> The reviewers all highlighted the privacy and ethical implications of this work, which we comment on here. We have been engaging with the ethical implications of our work in this domain throughout the process, both in the discriminative sense that the reviewers discuss (the privacy concerns), and in a generative sense (the ability to impersonate players). Because of this, we have worked with our institutional IRB to develop a protocol that has been approved prior to the paper for conducting research on these problems. Given this, we also agree that the discussion of these points in the paper was much too short, and didn’t accurately reflect the extent of ethical considerations that have gone into the research. We would plan to expand on this in the revised version.
>
> We agree that identification of individuals from behavioral traces raises significant privacy concerns, as well as concerns about power differentials and inequality, since it is often people who are marginalized in society who benefit most from anonymized communication and behavior. Taking this into account, it is useful to distinguish between two kinds of research into stylometry and de-anonymization. One form of this research, which we do not engage in, uses information or meta-data that is not generally publicly available to discover identities that would be infeasible to figure out without this type of data. A second form of this research uses data that is public and easily accessible to determine how distinguishable different individuals are from these public representations.
>
> This second form is the genre of stylometry research that we engage in — since all the chess data that we use is easily downloadable, and public by design — and we argue that this type of stylometry is important as part of a research program for understanding and addressing privacy risks in different domains. In particular, it is not a priori clear in any domain how uniquely identifying particular forms of behavioral or content data will be, but it is crucial to understand this, since it is easy to begin building individual-level classifiers from public data of this form, and individuals who operate in this domain with an expectation that they’ll remain anonymous may be relying on unrealistic assumptions about how revealing their data is. Moreover, there is a quantitative dimension to this – it is not simply all-or-nothing, but a question of how accurately these determinations can be made as a function of the data available.
>
> Only through a careful analysis of stylometry can we determine the level of risk in any given domain, and as such it is an important part of making people aware of privacy risks. And when public data is easily accessible, it is crucial to understand these risks. Our work is following this principle in illuminating the power of stylometry from behavioral traces, starting from an understanding of the risks involved. Finally, we note that we deliberately investigate and quantify these risks in the relatively harmless domain of chess, which we believe to be a valuable step before something similar is attempted in higher-stakes domains where more is at risk.
>
> ### Short points
>
> + We will update the abstract
> + The meaning of style will be added
> + We grouped by player numbers to match the previous work
> + We will try make it clear that D and R earlier in the paper
>
> ### _k_ value (also `iTpg`)
> In this paper, we use the convention that one move equals an action by both players (1 move = 2 ply). This indexing is natural for our work since our model takes sets of boards as input, but we will make the definition and comparison with previous work more explicit in the final version. Specifically, this means that our results are directly comparable with McIlroy-Young et al., as they conduct analyses on k=0, k=7.5, and k=15 using our indexing system, and we conduct analyses on k=0 and k=15. We will clarify this in the Table 3 caption.
>
> ### Narrower Rating Ranges
> The reviewer points out that our main analysis contains players of different skill levels, and asks how our method would fare on a version of the task restricted to players from a narrower range of skill levels. We tested this by restricting the range and domain to the 495 players rated between 1800-1900. The accuracy (k=15, seen and unseen) of our final model on this subset is 92.5%, which is higher than the comparable accuracy (85.4%) on the unrestricted set of players from all skill levels. This improvement suggests that the model is doing more than discriminating players by strength. We will include an extended version of this analysis in our final paper.
>
> ### Pretrained Initialization
> We looked into initializing the convolutional layers of our model with a pretrained model, but our input representation is significantly different (position pairs vs. 8 boards plus state information) from what Leela (open source AlphaZero implementation) uses. As our performance without pre-training was very high we didn't continue pursuing this direction, but we see it as a potentially promising avenue of future work.
>
> ### Loss Function (also `qhYv`)
> The reviewer rightfully points out that our loss function is optimized for differentiating individual players from each other. This is due to the task we focus on: behavioral stylometry in a low-risk domain. This does not necessarily mean that we cannot identify similar games or players: one can still measure the distance between game/player vectors in the embedding space. We will include such an example in our revision. Although the reviewer points out that some subsequences of moves are identical between games, these subsequences almost exclusively appear in the opening: after move k = 15, almost all sequences and subsequences are unique. Pretrained Initialization (a4DH). We looked into initializing the convolutional layers of our model with a pretrained model, but our input representation is significantly different (position pairs vs. 8 boards plus state information) from what Leela (open source _AlphaZero_ implementation) uses. As our performance without pre- training was very high we didn’t continue pursuing this direction, but we see it as a potentially promising avenue of future work.
>
> ### Visualization (also`RGQQ`)
> The visualizations shown in Figure 3 are based on models that are unable to see the first 15 moves of the game. By coloring the player vectors based on the opening moves chosen by the players, our intent was to show that the players’ embeddings are closely related to their opening preferences, and hence their stylistic preferences. Although this is easy to see in Fig. 3a, we agree that this is not easily discernible in 3b and 3c. This is largely due to the 2-D projection of the vectors: the clustering patterns are much clearer when using the interactive 3D viewer (_Tensorflow_ Embedding Projector). We included the necessary files to run this view locally in our supplement, and plan to host the viewer on a public facing server after the review period.

---

> > ### Comment · Reviewer_RGQQ · 2021-08-26
> > **Performance ceiling**
> >
> > One of a4DH's point is that whether this task has a performance ceiling -- which I don't think is addressed in your response.
> >
> > Chess, unlike games such as Go, has more rigid rules. As shown in the paper, it is much harder to identify top-performing players. If this is a meaningful task for few-shot learning, can you guys comment on where do you think the performance ceiling would be? It probably is a very difficult question and include lots of guesses. But as a4DH pointed out, it might be useful for the ML practitioners to know.

---

> > > ### Author Response · Authors · 2021-09-01
> > > **Performance Ceiling Author Comment**
> > >
> > > This is a good question that, as you mention, is quite difficult to answer with any degree of certainty at this stage, but we are happy to speculate. First, it is worth noting that maximum performance on this task will depend heavily on the specific parameters of the problem. For example, in our submission we achieve 54.0% accuracy with a set of 41,184 players and 85.4% on our main test set with 2,844 players. Accuracy is also affected by the number of games used (figure 2a explores this) so we would expect more games to increase accuracy. We find it plausible that the accuracy could be improved by future improvements; maybe a couple generations of optimizations could reach 80-90% on the 40,000 players set and approach 99% for the smaller sets. Since, anecdotally, even a couple of moves in a game can be enough to reveal a signature. We also hypothesize that the accuracy question might be answered by a formal study of the entropy of the moves of the players (i.e. consider each move to be a sample from a probability distribution that is unique to the player) allowing for bounds to be placed on the maximum accuracy.
> > >
> > > As you note one other component to consider is that not all populations of players are the same, higher skilled players are likely to be harder to identify as they tend to play the same moves as other high skilled players, i.e. make fewer mistakes. This is one of the factors behind our models not generalizing to the high-ranking players.
> > >
> > > We can also see our model being used to get a set of “nearby” candidates since we are learning an embedding with a distance metric. Or in a case like cheating, which a4DH mentioned, our model could be used to check if the accused player’s games are nearby their other games. Thus the model might not be used to identify the specific player, but to find a small set of candidates that could be checked with a more computationally intensive method.

---

> ### Author Response · Authors · 2021-09-04
> **Follow-up Question Check**
>
> The review discussion period is coming to an end. Are there any follow-up questions or points of clarification we can address before our ability to comment is turned off?

---

### Official Review · Reviewer_qhYv · 2021-07-17

**Rating:** 6
**Confidence:** 4

**Summary:**

In this paper the authors propose a transformer-based approach for capturing and identifying individual styles in chess. This goal, which the authors name Behavior Stylometry, is framed as an essential step towards algorithmic personalization and person-agent interaction. The suggested framework borrows ideas from advances in speech processing and only utilizes a small sample in order to learn a representation of players' actions. Once trained, the model can accurately identify the users in a larger pool of unseen samples. Lastly, the authors suggest that their approach reveals structural properties for characterizing playing style in chess.

**Ethical Concerns:**

As stated in the main review, one obvious ethical concern I have is that being able to identify users based on a small sample of their behavior, even in a context as seemingly innocuous as chess, may pose substantial privacy issues, which the authors themselves rightly note as well in the concluding section.

**Ethics Review Area:**

["Privacy and Security (e.g., consent)"]

**Limitations And Societal Impact:**

I believe the greatest limitation of this work is its specificity to the chess domain, making it harder to estimate how well the approach would generalize in other contexts, and therefore its broader impact.

**Main Review:**

Overall, this paper seems methodologically sound to me, and the results seem compelling enough. I do have certain qualms, however.

* The paper seems to be very much tailored for chess applications. While that makes the paper coherent and easy to follow, it also diminishes its impact. It is hard to estimate how much of the proposed approach would generalize to other domains, for instance, or how easy it would be to adapt the proposed framework for other tasks.

* The design of the loss function is meant to promote an embedding which keeps *individuals* as far apart in the embedding space as possible, but this approach seems to discount the fact that different people may have very similar playing styles, thus leading to artifacts in more general applications (I have more on that note, and how it relates to privacy concerns, see my last comment).

* From a training perspective, it is unclear how much intrinsic endogeneity may exist in the training data. That is to say, while you are separating games, entire subsequences within the game may be close to identical. To mitigate or at least study this aspect, it would make sense to cluster state-action subsequences within game trajectories and group/cluster them. How well would the model fare when seeing familiar people take slightly novel sequences?

* It would  have been interesting to see how well a naïve approach, such as comparing conditional action distributions, would have fared in this task.

* Lastly I'd like to point out that any approach for identifying users by a small sample of their actions may pose dire privacy concerns and should not be explored lightly. While I can see its utility in very restricted game-playing scenarios (for instance, opponent modeling), I believe it is a lot more valuable in typical contexts such as game-playing, recommendation etc. to characterize the *type* (indeed, the *style*) of a person or agent, rather than the unique identity of that person. Going for the latter goal seems to assume unnecessary privacy risk without adding much real-world utility to me.


**Needs Ethics Review:**

Yes

**Time Spent Reviewing:**

3

---

> ### Author Response · Authors · 2021-08-11
> **Author Response to Reviewer qhYv**
>
> Thank you for your thoughtful feedback and comments.
>
> We wrote a general statement on all the reviewer concerns regarding privacy, which is followed by comments specific to your review.
>
> ### Privacy (all)
> The reviewers all highlighted the privacy and ethical implications of this work, which we comment on here. We have been engaging with the ethical implications of our work in this domain throughout the process, both in the discriminative sense that the reviewers discuss (the privacy concerns), and in a generative sense (the ability to impersonate players). Because of this, we have worked with our institutional IRB to develop a protocol that has been approved prior to the paper for conducting research on these problems. Given this, we also agree that the discussion of these points in the paper was much too short, and didn’t accurately reflect the extent of ethical considerations that have gone into the research. We would plan to expand on this in the revised version.
>
> We agree that identification of individuals from behavioral traces raises significant privacy concerns, as well as concerns about power differentials and inequality, since it is often people who are marginalized in society who benefit most from anonymized communication and behavior. Taking this into account, it is useful to distinguish between two kinds of research into stylometry and de-anonymization. One form of this research, which we do not engage in, uses information or meta-data that is not generally publicly available to discover identities that would be infeasible to figure out without this type of data. A second form of this research uses data that is public and easily accessible to determine how distinguishable different individuals are from these public representations.
>
> This second form is the genre of stylometry research that we engage in — since all the chess data that we use is easily downloadable, and public by design — and we argue that this type of stylometry is important as part of a research program for understanding and addressing privacy risks in different domains. In particular, it is not a priori clear in any domain how uniquely identifying particular forms of behavioral or content data will be, but it is crucial to understand this, since it is easy to begin building individual-level classifiers from public data of this form, and individuals who operate in this domain with an expectation that they’ll remain anonymous may be relying on unrealistic assumptions about how revealing their data is. Moreover, there is a quantitative dimension to this – it is not simply all-or-nothing, but a question of how accurately these determinations can be made as a function of the data available.
>
> Only through a careful analysis of stylometry can we determine the level of risk in any given domain, and as such it is an important part of making people aware of privacy risks. And when public data is easily accessible, it is crucial to understand these risks. Our work is following this principle in illuminating the power of stylometry from behavioral traces, starting from an understanding of the risks involved. Finally, we note that we deliberately investigate and quantify these risks in the relatively harmless domain of chess, which we believe to be a valuable step before something similar is attempted in higher-stakes domains where more is at risk.
>
> ### Generalization
> We propose chess as a model system in which we introduce the problem of behavioral stylometry and demonstrate that a technique from another domain (human speech synthesis) does very well. Thus the method has generalized from at least one other domain, and we see chess as a natural testbed to explore both how identifiable people are from their actions and the benefits versus risks of behavioral stylometry in a relatively low-risk domain. We hope the insights we gain here will provide insight into applying these methods to other domains in future work.
>
> ### Loss Function (also `a4DH`)
> The reviewer rightfully points out that our loss function is optimized for differentiating individual players from each other. This is due to the task we focus on: behavioral stylometry in a low-risk domain. This does not necessarily mean that we cannot identify similar games or players: one can still measure the distance between game/player vectors in the embedding space. We will include such an example in our revision. Although the reviewer points out that some subsequences of moves are identical between games, these subsequences almost exclusively appear in the opening: after move k = 15, almost all sequences and subsequences are unique. Pretrained Initialization (a4DH). We looked into initializing the convolutional layers of our model with a pretrained model, but our input representation is significantly different (position pairs vs. 8 boards plus state information) from what Leela (open source _AlphaZero_ implementation) uses. As our performance without pre- training was very high we didn’t continue pursuing this direction, but we see it as a potentially promising avenue of future work.
>
> ### Subsequences
> Identical subsequnces in a chess game are rare, outside of the opening game. Most positions encountered during a game are unique to that game, figure 3 of McIlroy-Young et al. 2020b shows how rare they are, with 0.02% of the positions in the training set seen in the test set. Looking for similar positions instead of identical is also difficult as we are not aware of a good similarity measures for this type of task, and in fact was one of the motivations for this work. So answering the reviewers question is a good topic for future works.
>
> ### Conditional action distribution
> The reviewer made the useful suggestion of comparing against a naïve baseline using conditional action distributions. In response to this, we implemented the following baseline: for each user, we computde their empirical distribution over the 4,096 possible actions for all their moves in their support set of of games (starting with move k=15 for direct comparison with other models). We then treated these as vectors, as in our main model, and compared their similarity for identifying players. This resulted in an accuracy of 27.7% (compared with 24.4% for our 5-opening-moves baseline) on the 10k+ task. We will include a full analysis in the final paper.

---

> > ### Comment · Reviewer_qhYv · 2021-09-01
> > **Response to authors**
> >
> > Thank you for the thoughtful reply and apologies for the belated response, it's been a difficult few weeks.
> >
> > Regarding the privacy issue, while it's true that understanding how individuals can be uniquely identified in publicly available datasets is an important research question, and separate from that of actively identifying users using data that is not publicly available, I'm not sure how well this distinction serves to assuage privacy concerns. It is easy to see how in the process of studying how to identify users in publically available data, one is actively discovering the means for malicious actors to do so. That being said, since this *is* an important research topic, space and time permitting, it would be beneficial to include in the paper an analysis or at least some discussion of what means one might employ to avoid being identified.
> >
> > Regarding generalization, I'm not sure it's compelling to argue that since the techniques used for chess stylometry originate from speech synthesis, the general approach (i.e. unique style identification) would be directly relevant, let alone applicable, in other domains, but it is true that in itself showing that techniques from the world of speech synthesis are applicable in a surprisingly different domain of unique style identification in chess holds merit. However, in their response regarding the loss function, the authors comment that after move k=15 almost all subsequences are unique seems profoundly unique to chess, or at least to the categories of games very similar to chess, which yet again raises concerns on how well the approach would learn to separate very similar actors in other domains, where this key characteristic in the data is unlikely to manifest itself.
> >
> > Having said all that, I thank the authors again for the their detailed response and the work they put into it (as well as the work they intend to add to the submission, which I believe would strengthen the paper). I stand by my assessment that this is solid work.

---

> > > ### Author Response · Authors · 2021-09-04
> > > **Author Comment on Response**
> > >
> > > Thank you for your time in reading our response and continuing the discussion. Your suggestion to include an analysis/discussion of what means one might employ to avoid identification is a great one, and we’ll certainly include this.
> > >
> > > As for the uniqueness of sequences of actions, we believe this to be quite general and not limited to games such as chess. In chess, there are a relatively few number of actions one can take in any given position, but the combinatorial nature of chess means that most positions encountered are unique (and hence most subsequences of positions are unique). In other domains, a similar complexity emerges; for example, in conversation, there are a relatively small number of words a typical person will say in any given moment, but most conversations quickly reach a unique state (and hence most subsequences of conversations are unique). In robotics, the action set is relatively small but the combinatorial explosion over the course of an interaction or task results in unique situations as well.
> > >
> > > Thanks again for your time and input.

---

### Official Review · Reviewer_RGQQ · 2021-07-18

**Rating:** 7
**Confidence:** 4

**Summary:**

This paper introduces a task of behavioral stylometry in the domain of chess plays. It uses a few-shot learning framework and uses the optimization objective inspired by the speaker identification domain to solve this problem. It demonstrated clear gains over previous baselines and has performed a satisfactory analysis that can be used by future researchers to extend this work.

**Limitations And Societal Impact:**

The authors have addressed other limitations well.

**Main Review:**

**Originality**: this work opens up a new and meaningful task domain for few-shot learning and representation learning. Though the framework and optimization objective are heavily inspired by other domain, the application is very novel.

**Quality**: the quality of this paper is very high. The authors have conducted satisfactory experiments to evaluate the model's performance. The original experiment shows the promise of using few-shot learning as a framework to solve this task. The ablation studies (by whether to include first k moves) and evaluation of top players point the research community to future research directions.

**Clarity**: the structure and writing are very clear.

**Significance**: it is very important to introduce new problems to the machine learning community. Online chess communities have been growing for quite a few years, and this domain has a strong connection to human behavior modeling, teaching, and human-AI collaboration. It is left to the ML community to decide what are interesting problems that ML should solve in the domain of chess play. The authors of this paper built upon prior work and defined player identification as one of the important tasks to solve -- embeddings that capture the behavioral style of play are effective embeddings for chess moves. This has a huge implication for unsupervised learning and embedding evaluations in the future. It might not be THE task or THE evaluation metric that everyone needs to care about, but this is a very solid first step.

I don't have any significant issues with this paper, mainly because this paper introduces a new problem in a new domain. This is not to say that the paper is perfect. For example:
1. The paper proposes to build "embeddings" of player style, but the visualization in Figure 3 (b) (c), I can't easily recognize clusters, making me wonder if the embeddings do contain valuable information beyond player identification.
2. It seems that the data is public but the player IDs can only be obtained through private communication. I'm concerned that this level of privacy will impact future researchers using this as a valid task to test new algorithms. I hope the authors of this paper will spend more time thinking about how to build a research community around this domain and how to enable rapid model development in this domain in the future.


**Time Spent Reviewing:**

3

---

> ### Author Response · Authors · 2021-08-11
> **Response to Reviewer RGQQ**
>
> Thank you for your thoughtful feedback and comments.
>
> We wrote a general statement on all the reviewer concerns regarding privacy (the other reviewers have important concerns), which is followed by comments specific to your review.
>
> ### Privacy (all)
>
> The reviewers all highlighted the privacy and ethical implications of this work, which we comment on here. We have been engaging with the ethical implications of our work in this domain throughout the process, both in the discriminative sense that the reviewers discuss (the privacy concerns), and in a generative sense (the ability to impersonate players). Because of this, we have worked with our institutional IRB to develop a protocol that has been approved prior to the paper for conducting research on these problems. Given this, we also agree that the discussion of these points in the paper was much too short, and didn’t accurately reflect the extent of ethical considerations that have gone into the research. We would plan to expand on this in the revised version.
>
> We agree that identification of individuals from behavioral traces raises significant privacy concerns, as well as concerns about power differentials and inequality, since it is often people who are marginalized in society who benefit most from anonymized communication and behavior. Taking this into account, it is useful to distinguish between two kinds of research into stylometry and de-anonymization. One form of this research, which we do not engage in, uses information or meta-data that is not generally publicly available to discover identities that would be infeasible to figure out without this type of data. A second form of this research uses data that is public and easily accessible to determine how distinguishable different individuals are from these public representations.
>
> This second form is the genre of stylometry research that we engage in — since all the chess data that we use is easily downloadable, and public by design — and we argue that this type of stylometry is important as part of a research program for understanding and addressing privacy risks in different domains. In particular, it is not a priori clear in any domain how uniquely identifying particular forms of behavioral or content data will be, but it is crucial to understand this, since it is easy to begin building individual-level classifiers from public data of this form, and individuals who operate in this domain with an expectation that they’ll remain anonymous may be relying on unrealistic assumptions about how revealing their data is. Moreover, there is a quantitative dimension to this – it is not simply all-or-nothing, but a question of how accurately these determinations can be made as a function of the data available.
>
> Only through a careful analysis of stylometry can we determine the level of risk in any given domain, and as such it is an important part of making people aware of privacy risks. And when public data is easily accessible, it is crucial to understand these risks. Our work is following this principle in illuminating the power of stylometry from behavioral traces, starting from an understanding of the risks involved. Finally, we note that we deliberately investigate and quantify these risks in the relatively harmless domain of chess, which we believe to be a valuable step before something similar is attempted in higher-stakes domains where more is at risk.
>
> ### Reproducibility
> We agree that balancing privacy concerns with reproducibility is a delicate challenge. We are weighing the pros and cons of various release options, such as restricting release to other researchers only or anonymizing games and player IDs. We plan to ensure that we can build a research community around using chess as a model system for behavioral stylometry and related goals while respecting the chess players who provide their data.
>
> The original submission included a misleading reference to personal communication in footnote 1 that we will fix in the final version of the paper — we did not intend for this to imply that the data is not public. In fact, all of our results can be replicated on data that is completely public, and the personal communication was only to match particular subsets of player IDs from McIlroy-Young et al. 2020b. (But the results would have been the same had we used different subsets of players with the same properties.)
>
> ### Visualization (also `a4DH`)
> The visualizations shown in Figure 3 are based on models that are unable to see the first 15 moves of the game. By coloring the player vectors based on the opening moves chosen by the players, our intent was to show that the players’ embeddings are closely related to their opening preferences, and hence their stylistic preferences. Although this is easy to see in Fig. 3a, we agree that this is not easily discernible in 3b and 3c. This is largely due to the 2-D projection of the vectors: the clustering patterns are much clearer when using the interactive 3D viewer (_Tensorflow_ Embedding Projector). We included the necessary files to run this view locally in our supplement, and plan to host the viewer on a public facing server after the review period.

---

### Review · Ethics_Reviewer_WYHD · 2021-08-10

**Recommendation:**

In the end, the paper should do a better job acknowledging how this research domain could be applied in ways that threaten people's privacy and autonomy. While there might be benign applications, it almost certainly would be attractive to organizations and state actors who seek to identify people.

**Ethical Issues:**

Yes

**Ethics Review:**

The paper develops new means of behavioral stylometry. While the research is taking place in a relatively harmless domain (chess-playing style) and the researchers aren't attempting to directly identify the subjects in the experimental data (and this dataset doesn't contain any obvious PII), there are clearly broader social implications about the potential for identifying people simply based on some behavioral data. The authors give only a passing mention to the fact this area of research might have privacy and surveillance implications.

---

> ### Author Response · Authors · 2021-08-30
> **Authors Response to Ethics Reviewer WYHD**
>
> Thank you for your insightful comments, we agree and will add more discussion of the privacy concerns to the final paper.
>
> We addressed some of your points in our general author response (posted after your comments) and our last paragraph of the privacy section is relevant to your points:
>
> > Only through a careful analysis of stylometry can we determine the level of risk in any given domain, and as such it is an important part of making people aware of privacy risks. And when public data is easily accessible, it is crucial to understand these risks. Our work is following this principle in illuminating the power of stylometry from behavioral traces, starting from an understanding of the risks involved. Finally, we note that we deliberately investigate and quantify these risks in the relatively harmless domain of chess, which we believe to be a valuable step before something similar is attempted in higher-stakes domains where more is at risk.
>
> In addition we would like to clarify the future goals of removing privacy risks: Our overarching goal that this project is part of is to build systems that can understand players allowing them to provide useful feedback, find teachable mistakes, give useful examples to learn from. Identification of people is our focus of this paper since it is something that can be optimized for directly. The last plot of the paper figure 3 was included to show that the methods do more than identify people, they also can be used to find similar people/games.
>
> In retrospect, we should have incorporated the above ethical discussion into the submission itself. We overlooked the connection between the extra allowed page and the lack of a broader impact section -- had we realized this, we would have certainly devoted a page to discussing the various ethical and societal issues raised by our work. Addressing this is our top priority for the revision.

---

### Review · Ethics_Reviewer_hTQw · 2021-08-12

**Recommendation:**

It is possible to address them in the current version by expanding discussion on the risks of this type of technique along with more explanation of why the domain of chess is more benign than other domains.

**Ethical Issues:**

Yes

**Ethics Review:**

The authors do briefly note the main ethical challenge the paper raises: the risk that this technique will be used to identify people who wish to remain private.  Indeed, the ability to conduct few shot identification of individuals based on their decisions is likely to be of interest to marketers, law enforcement, and many other entities in ways that might give rise to privacy violations and abuse.  The authors note that the example of publicly presented chess plays is far more benign than other uses, but further discussion of this in the paper itself would be valuable, along with an acknowledgment that these same techniques will likely be abused as they continue to improve.

---

> ### Author Response · Authors · 2021-09-01
> **Author Response to Ethics Reviewer hTQw**
>
> Thank you for your thoughtful comments.
>
> Our top priority for the revision is to add more discussion of the privacy implications of this work. As we write in our response to the original reviewers:
>
> We have been engaging with the ethical implications of our work in this domain throughout the process, both in the discriminative sense that the reviewers discuss (the privacy concerns), and in a generative sense (the ability to impersonate players). Because of this, we have worked with our institutional IRB to develop a protocol that has been approved prior to the paper for conducting research on these problems. Given this, we also agree that the discussion of these points in the paper was much too short, and didn’t accurately reflect the extent of ethical considerations that have gone into the research. We would plan to expand on this in the revised version.
>
> We agree that identification of individuals from behavioral traces raises significant privacy concerns, as well as concerns about power differentials and inequality, since it is often people who are marginalized in society who benefit most from anonymized communication and behavior. Taking this into account, it is useful to distinguish between two kinds of research into stylometry and de-anonymization. One form of this research, which we do not engage in, uses information or meta-data that is not generally publicly available to discover identities that would be infeasible to figure out without this type of data. A second form of this research uses data that is public and easily accessible to determine how distinguishable different individuals are from these public representations.
> This second form is the genre of stylometry research that we engage in — since all the chess data that we use is easily downloadable, and public by design — and we argue that this type of stylometry is important as part of a research program for understanding and addressing privacy risks in different domains. In particular, it is not a priori clear in any domain how uniquely identifying particular forms of behavioral or content data will be, but it is crucial to understand this, since it is easy to begin building individual-level classifiers from public data of this form, and individuals who operate in this domain with an expectation that they’ll remain anonymous may be relying on unrealistic assumptions about how revealing their data is. Moreover, there is a quantitative dimension to this – it is not simply all-or-nothing, but a question of how accurately these determinations can be made as a function of the data available.
> Only through a careful analysis of stylometry can we determine the level of risk in any given domain, and as such it is an important part of making people aware of privacy risks. And when public data is easily accessible, it is crucial to understand these risks. Our work is following this principle in illuminating the power of stylometry from behavioral traces, starting from an understanding of the risks involved. Finally, we note that we deliberately investigate and quantify these risks in the relatively harmless domain of chess, which we believe to be a valuable step before something similar is attempted in higher-stakes domains where more is at risk.
>
> We will follow your suggestion of adding further discussion of why chess is a more benign domain than others, as well as acknowledge and discuss the possibility of abuse in other domains as this type of technique improves. We believe that chess is a more benign domain compared to others such as text, spoken language, or online behaviour for multiple reasons. First the data are public by design—e.g. online chess platforms make it easy for anyone to view past games by any other player, and thus we expect most players to be aware of the public nature of their games. Second, by default online chess profiles contain essentially no other information about the user besides the games they play and their chess ratings. Thus, in comparison with a domain such as author identification of text, where de-identification could link an individual with opinions, beliefs, and other expressions, in chess de-identification only links an individual with their chess moves (which isn’t devoid of harm, but we believe to be more benign).
>
> In retrospect, we should have incorporated the above ethical discussion into the submission itself. We overlooked the connection between the extra allowed page and the lack of a broader impact section -- had we realized this, we would have certainly devoted a page to discussing the various ethical and societal issues raised by our work. Addressing this is our top priority for the revision.

---

### Author Response · Authors · 2021-08-11
**Author Response**

We would like to thank the reviewers for their thoughtful feedback, and address their concerns here.

This is the full set of responses and is a copy of our response to the individual reviewers.

### Privacy (all)
The reviewers all highlighted the privacy and ethical implications of this work, which we comment on here. We have been engaging with the ethical implications of our work in this domain throughout the process, both in the discriminative sense that the reviewers discuss (the privacy concerns), and in a generative sense (the ability to impersonate players). Because of this, we have worked with our institutional IRB to develop a protocol that has been approved prior to the paper for conducting research on these problems. Given this, we also agree that the discussion of these points in the paper was much too short, and didn’t accurately reflect the extent of ethical considerations that have gone into the research. We would plan to expand on this in the revised version.

We agree that identification of individuals from behavioral traces raises significant privacy concerns, as well as concerns about power differentials and inequality, since it is often people who are marginalized in society who benefit most from anonymized communication and behavior. Taking this into account, it is useful to distinguish between two kinds of research into stylometry and de-anonymization. One form of this research, which we do not engage in, uses information or meta-data that is not generally publicly available to discover identities that would be infeasible to figure out without this type of data. A second form of this research uses data that is public and easily accessible to determine how distinguishable different individuals are from these public representations.

This second form is the genre of stylometry research that we engage in — since all the chess data that we use is easily downloadable, and public by design — and we argue that this type of stylometry is important as part of a research program for understanding and addressing privacy risks in different domains. In particular, it is not a priori clear in any domain how uniquely identifying particular forms of behavioral or content data will be, but it is crucial to understand this, since it is easy to begin building individual-level classifiers from public data of this form, and individuals who operate in this domain with an expectation that they’ll remain anonymous may be relying on unrealistic assumptions about how revealing their data is. Moreover, there is a quantitative dimension to this – it is not simply all-or-nothing, but a question of how accurately these determinations can be made as a function of the data available.

Only through a careful analysis of stylometry can we determine the level of risk in any given domain, and as such it is an important part of making people aware of privacy risks. And when public data is easily accessible, it is crucial to understand these risks. Our work is following this principle in illuminating the power of stylometry from behavioral traces, starting from an understanding of the risks involved. Finally, we note that we deliberately investigate and quantify these risks in the relatively harmless domain of chess, which we believe to be a valuable step before something similar is attempted in higher-stakes domains where more is at risk.

### Reproducibility (`RGQQ`)
We agree that balancing privacy concerns with reproducibility is a delicate challenge. We are weighing the pros and cons of various release options, such as restricting release to other researchers only or anonymizing games and player IDs. We plan to ensure that we can build a research community around using chess as a model system for behavioral stylometry and related goals while respecting the chess players who provide their data.

The original submission included a misleading reference to personal communication in footnote 1 that we will fix in the final version of the paper — we did not intend for this to imply that the data is not public. In fact, all of our results can be replicated on data that is completely public, and the personal communication was only to match particular subsets of player IDs from McIlroy-Young et al. 2020b. (But the results would have been the same had we used different subsets of players with the same properties.)

### _k_ value (`iTpg`, `a4DH`)
In this paper, we use the convention that one move equals an action by both players (1 move = 2 ply). This indexing is natural for our work since our model takes sets of boards as input, but we will make the definition and comparison with previous work more explicit in the final version. Specifically, this means that our results are directly comparable with McIlroy-Young et al., as they conduct analyses on k=0, k=7.5, and k=15 using our indexing system, and we conduct analyses on k=0 and k=15. We will clarify this in the Table 3 caption.

### Model Choice (`iTpg`)
We hypothesized that the use of a transformer architecture would improve performance because the task depends on decisions made throughout the game, with some presumably being much more important than others. The self-attention mechanism is designed to allow the model to use any past state easily, while models such as LSTMs use a hidden state vector to indirectly incorporate past states in their outputs. We also tested the performance of both and found transformers had lower validation error and loss during our model tuning. As an additional test we have trained a bi-directional LSTM model using the full dataset and it achieves 78.0% accuracy on the 10k+ task with k=15, which is lower than the transformer model which achieves 86.0% on the same task. The loss and error on validation set are also worse when compared with a bi-directional LSTM model. The equal error rate for the models was: 0.225 LSTM and 0.187 transformer, and the loss was 2.502 LSTM and 2.323 transformer. Conditional action distribution (qhYv). The reviewer made the useful suggestion of comparing against a naïve baseline using conditional action distributions. In response to this, we implemented the following baseline: for each user, we computde their empirical distribution over the 4,096 possible actions for all their moves in their support set of of games (starting with move k=15 for direct comparison with other models). We then treated these as vectors, as in our main model, and compared their similarity for identifying players. This resulted in an accuracy of 27.7% (compared with 24.4% for our 5-opening-moves baseline) on the 10k+ task. We will include a full analysis in the final paper.

### Narrower Rating Ranges (`a4DH`)
The reviewer points out that our main analysis contains players of different skill levels, and asks how our method would fare on a version of the task restricted to players from a narrower range of skill levels. We tested this by restricting the range and domain to the 495 players rated between 1800-1900. The accuracy (k=15, seen and unseen) of our final model on this subset is 92.5%, which is higher than the comparable accuracy (85.4%) on the unrestricted set of players from all skill levels. This improvement suggests that the model is doing more than discriminating players by strength. We will include an extended version of this analysis in our final paper.

### Generalization (`qhYv`)
We propose chess as a model system in which we introduce the problem of behavioral stylometry and demonstrate that a technique from another domain (human speech synthesis) does very well. Thus the method has generalized from at least one other domain, and we see chess as a natural testbed to explore both how identifiable people are from their actions and the benefits versus risks of behavioral stylometry in a relatively low-risk domain. We hope the insights we gain here will provide insight into applying these methods to other domains in future work.

### Loss Function (`qhYv`, `a4DH`)
The reviewer rightfully points out that our loss function is optimized for differentiating individual players from each other. This is due to the task we focus on: behavioral stylometry in a low-risk domain. This does not necessarily mean that we cannot identify similar games or players: one can still measure the distance between game/player vectors in the embedding space. We will include such an example in our revision. Although the reviewer points out that some subsequences of moves are identical between games, these subsequences almost exclusively appear in the opening: after move k = 15, almost all sequences and subsequences are unique. Pretrained Initialization (a4DH). We looked into initializing the convolutional layers of our model with a pretrained model, but our input representation is significantly different (position pairs vs. 8 boards plus state information) from what Leela (open source _AlphaZero_ implementation) uses. As our performance without pre- training was very high we didn’t continue pursuing this direction, but we see it as a potentially promising avenue of future work.

### Visualization (`RGQQ`, `a4DH`)
The visualizations shown in Figure 3 are based on models that are unable to see the first 15 moves of the game. By coloring the player vectors based on the opening moves chosen by the players, our intent was to show that the players’ embeddings are closely related to their opening preferences, and hence their stylistic preferences. Although this is easy to see in Fig. 3a, we agree that this is not easily discernible in 3b and 3c. This is largely due to the 2-D projection of the vectors: the clustering patterns are much clearer when using the interactive 3D viewer (_Tensorflow_ Embedding Projector). We included the necessary files to run this view locally in our supplement, and plan to host the viewer on a public facing server after the review period.

---

### Decision · Program_Chairs · 2021-09-27

**Decision:**

Accept (Poster)

**Comment:**

UPDATE:  The revision has been reviewed and this paper has been accepted.  We recommend that the authors integrate the final sentence of the introduction into the end of Paragraph 2, where they are talking about applications of their ideas.  Some applications raise ethical considerations and it's important to admit that alongside the other applications rather than it appearing as an afterthought at the end.

----

This paper is very difficult to make a decision.  The reviewers all agree the paper is excellent technically (even the low reviewer score is not basing their score on the technical contribution).  However, there is also general agreement that the paper does not sufficiently address what amounts to serious ethical considerations of this work, specifically the opportunity for it or derivatives to be abused.  The authors acknowledge that a more substantial discussion is needed in the paper, so I trust they understand this significance of this concern.

However, I (and other reviewers) also feel the authors' response lacks a recognition of appropriate mitigations of possible abuse.  They primarily appeal to the public nature of the chess data as a consideration in why this line of research is ethically appropriate.  This is not sufficient.  Abuses, for example where a bad actor uses even public data to identify/track an otherwise anonymous person in order to discriminate or restrict rights, are not limited to use only private data.  In fact, it seems to me public data abuses are more likely or more damaging than private data abuses (as the latter limits the abuse to owners of private data).

I do believe advances of stylometry in identifying chess players is a pretty benign application, however there's little discussion about how general this work might be in its ability to do stylometry in other domains.  The authors mention an additional social benefit of understanding what identifiability exists in our datasets, but this argument needs more attention and peer review than can be completed through the author response/discussion.

As a consequence this paper is being **conditionally accepted**.  The final determination will only be made upon review of the revision.   The authors should take advantage of the extra page to address concerns as reflected in their own response(s), although without the emphasis that use of public data lessens the ethical concerns.  We hope that the authors' added discussion could be used to send a strong signal commensurate with the gravity of the situation, so that appropriate community norms can be developed.